# Proper migration of lymphatic endothelial cells requires survival and guidance cues from arterial mural cells

**Di Peng[1], Koji Ando[2]\*, Melina Hußmann[3], Marleen Gloger[1], Renae Skoczylas[1], Naoki Mochizuki[4], Christer Betsholtz[5,6], Shigetomo Fukuhara[2], Stefan Schulte-Merker[3], Nathan D Lawson[7], Katarzyna Koltowska[1]\***

[1]Uppsala University, Immunology Genetics and Pathology, Uppsala, Sweden; [2]Department of Molecular Pathophysiology, Institute of Advanced Medical Sciences, Nippon Medical School, Tokyo, Japan; [3]Institute of Cardiovascular Organogenesis and Regeneration, Faculty of Medicine, WWU Münster, Münster, Germany; [4]Department of Cell Biology, National Cerebral and Cardiovascular Center Research Institute, Suita, Japan; [5]Department of Immunology, Genetics and Pathology, Rudbeck Laboratory, Uppsala University, Uppsala, Sweden; [6]Department of Medicine Huddinge (MedH), Karolinska Institutet, Campus Flemingsberg, Huddinge, Sweden; [7]Department of Molecular, Cellular, and Cancer Biology, University of Massachusetts Medical School, Worcester, United States

**\*For correspondence:**
koji-ando@nms.ac.jp (KA);
kaska.koltowska@igp.uu.se (KK)

**Competing interest:** The authors declare that no competing interests exist.

**Abstract** The migration of lymphatic endothelial cells (LECs) is key for the development of the complex and vast lymphatic vascular network that pervades most tissues in an organism. In zebrafish, arterial intersegmental vessels together with chemokines have been shown to promote lymphatic cell migration from the horizontal myoseptum (HM). We observed that emergence of mural cells around the intersegmental arteries coincides with lymphatic departure from HM which raised the possibility that arterial mural cells promote LEC migration. Our live imaging and cell ablation experiments revealed that LECs migrate slower and fail to establish the lymphatic vascular network in the absence of arterial mural cells. We determined that mural cells are a source for the C-X-C motif chemokine 12 (Cxcl12a and Cxcl12b), vascular endothelial growth factor C (Vegfc) and collagen and calcium-binding EGF domain-containing protein 1 (Ccbe1). We showed that chemokine and growth factor signalling function cooperatively to induce robust LEC migration. Specifically, Vegfc-Vegfr3 signalling, but not chemokines, induces extracellular signal-regulated kinase (ERK) activation in LECs, and has an additional pro-survival role in LECs during the migration. Together, the identification of mural cells as a source for signals that guide LEC migration and survival will be important in the future design for rebuilding lymphatic vessels in disease contexts.

## Editor's evaluation

Utilizing the zebrafish models and the state-of-art approaches, this study clearly identifies that the arterial mural cells serve as the source for the chemokines and growth factors, which are key factors not only for the migration and survival of lymphatic endothelial cells but also for the building of lymphatic networks in most organs during embryonic development. The findings and conclusion would be an important platform in the future design for rebuilding lymphatic vessels in treating lymphatic-deficient diseases including lymphedema.

## Introduction

The lymphatic vessel network spans across the whole body to balance tissue fluid homeostasis, coordinate the immune responses, and enable dietary fat absorption in the intestine. The robustness of the formation and reproducibility of the vascular tree is dependent on molecular dynamics and tissue-tissue interaction required for the precision and fine-tuning of lymphatic endothelial cell (LEC) migration. Although multiple previous studies have uncovered important signals and cells guiding lymphatic vessel formation (*Bussmann et al., 2010*; *Cha et al., 2012*; *Jafree et al., 2021*), recent technological developments and new transgenic lines (*Ando et al., 2016*; *Wang et al., 2020*) have opened up opportunities to identify further regulators of LEC migration.

VEGFC-mediated signalling through the vascular endothelial growth factor receptor 3 (VEGFR3) is essential for multiple steps of lymphatic vessels formation, including LEC proliferation, differentiation, and migration. In vitro cell culture experiments have demonstrated that VEGFC-VEGFR3 and β1 integrin to promote LEC migration (*Mäkinen et al., 2001*; *Wang et al., 2001*). Studies using a *vegfc* zebrafish reporter line have uncovered multiple sources of *vegfc*, including fibroblasts and neurons, which contribute to the initial sprouting and migration of lymphatic vessel into the HM (early migration) (*Wang et al., 2020*). The requirement of Vegfc-Vegfr3 and its source(s) in the LEC migration out of the HM (late migration) remains to be defined. Mechanistically, transcription factor MAFB, which regulates LEC migration but not proliferation, has been shown to act downstream of VEGFC-VEGFR3 signalling (*Dieterich et al., 2015*; *Koltowska et al., 2015b*). A genome-wide analysis further indicated the presence of a transcriptional network controlling LEC migration, through the induction of chemokine receptors that promote chemotaxis in migrating LECs (*Williams et al., 2017*). Although migratory regulators have been identified, the upstream cellular source of the signals initiating the migration is unknown.

In zebrafish trunk, lymphatic vessel specification is marked by the expression of transcription factor Prox1 in ECs around 32 hours post fertilization (hpf) in response to Vegfc-Vegfr3 signalling (*Koltowska et al., 2015a*). Around 34 hpf, venous-derived Prox1-positive cells sprout from the posterior cardinal vein (PCV) and migrate to the horizontal myoseptum (HM), establishing parachordal lymphatic endothelial cells (PLs) (*Hogan and Schulte-Merker, 2017*). After about 10 hours the PLs move out from the HM region and migrate dorsally or ventrally, and by 5 days post fertilization (dpf) give raise to the main trunk lymphatic vessels, including dorsal longitudinal lymphatic vessel (DLLV), intersegmental lymphatic vessel (ISLV), and thoracic duct (TD) (*Küchler et al., 2006*; *Yaniv et al., 2006*). During this later migration, the vast majority of the LECs are associated with the arterial intersegmental vessels (aISVs) (*Bussmann et al., 2010*). Given that PLs remain in the HM, and subsequently the lymphatic network formation is compromised in mutant embryos lacking aISVs (*plcy*[t26480] and *kdrl*[hu5088] mutants), the aISVs are instrumental in LEC migration (*Bussmann et al., 2010*). On the molecular level, arterial ECs (aECs) reportedly secrete Cxcl12b to guide this LECs migration via the Cxcr4 receptor expressed in LECs (*Cha et al., 2012*). Yet, whether other tissues or cells cooperate with aECs to support this migration remains unknown. Simultaneous with LEC development described above, vascular mural cells (MCs) are formed de novo along aISVs and beneath the dorsal aorta. aISVs play a critical role in this process, and MC emergence is completely abolished in the absence of aECs (*Ando et al., 2016*). The spatio-temporal similarity of MC and lymphatic vessel development around aISVs raises the question about a possible interaction between MCs and LECs in this region.

Here, we took advantage of transgenic zebrafish reporters which allowed us to visualize MCs and LECs simultaneously at high spatio-temporal resolution in vivo, and to investigate their communication during lymphangiogenesis. We found that MCs emergence precedes LEC migration along aISV and that LECs interact with MCs residing at the aISVs. Moreover, in the absence of MCs, LEC migration was inhibited, and lymphatic vessel formation was compromised. We further determined that MCs produce lymphangiogenic factors including *vegfc*, *cxcl12a*, and *cxcl12b*. Thus, this study uncovers a close interaction between MC and LEC, which is of a functional importance for lymphatic vessel formation in the zebrafish trunk.

## Results

### MCs and LECs interact during LEC migration

To address a potential interaction between MCs and LECs around arteries, we examined their distribution around aISVs, using the reporter lines *Tg(lyve1b:DsRed);Tg(flt1:YFP);Tg(pdgfrb:GFP),* where *lyve* labels veins and lymphatics, *flt1* arteries and *pdgfrb* high expression the MCs (*Figure 1A–B* and *Figure 1—figure supplement 1*). We found the spatial proximity, with MCs being sandwiched between the aISV and the migrating LEC at 4 dpf (*Figure 1A*). Subsequently, to identify the temporal sequence of LEC migration and appearance of MC along intersegmental vessels, we performed time-lapse imaging using the above reporter lines. We have observed that LECs migrated out from HM immediately after the emergence of *pdgfrb*+ MCs (*Figure 1B and C*, *Figure 1—video 1*). We confirmed these observations by time-lapse imaging of MCs and LECs, and blood vessels, respectively, in *Tg(lyve-1b:DsRed);Tg(kdrl:TagBFP);Tg(pdgfrb:GFP)* transgenic lines that allows separation of lymphatic, only labelled by *lyve,* from veins which are co-labelled by *lyve* and *kdrl* (*Figure 1—figure supplement 1*, *Figure 1—video 2*), and in *Tg(dab2:GAL4FF);Tg(UAS:GFP);Tg(pdgfrb:mCherry)* where *dab2* is expressed in LECs and venous endothelial cells (*Figure 1—figure supplement 1*, *Figure 1—video 3*). Utilizing these reporters, we found that in approximately 90% of the cases, LEC migrated towards and interacted with the MC residing on aISV (n = 21 *Figure 1D–E*, *Figure 1—video 5*, *Figure 1—video 6*). The number of MCs was not changed before and after the LEC migration (*Figure 1*). When LECs migrated out from the HM region, we noticed that LECs dynamically extended and regressed protrusions and actively reach towards the MCs (*Figure 1E*, *Figure 1—video 4*), while MCs appeared still on aISV (*Figure 1—figure supplement 1*). To understand the biological significance of the interaction between LECs and MCs, we quantified the velocity of LEC migration along aISV and found that the LECs in contact with the MCs migrated two times faster than LECs migrating along aISV without MCs (*Figure 1G*). These observations suggest that MCs might provide directional cues to promote robust LEC migration.

### MCs promote lymphatic vessel formation

We next asked if *pdgfrb*-positive MCs are necessary for lymphatic vessels formation. PDGFRβ is known to be essential for MC development, especially their proliferation and migration (*Ando et al., 2016*; *Gaengel et al., 2009*). The *pdgfrb*^um148^ mutant zebrafish (*Kok et al., 2015*) showed a 30% reduction of MC number around aISVs (*Figure 1H–I*). Coincidently, trunk lymphatic vasculature formation in *pdgfrb*^um148^ mutant zebrafish revealed slight reduction in the network formation. The rendering of *lyve:DsRed* labelled lymphatic vessels, as a measurement of lymphatic vessels density, revealed on average an area of 1 mm$^2$ in the sibling vs. 0.7 mm$^2$ in the *pdgfrb*^um148^ mutants (*Figure 1H and J*). Treatment with a PDGFRβ inhibitor, AG1296 from 48 hpf onwards, led to a greater reduction in MC coverage and LECs number (*Figure 1—figure supplement 2A-B*). Together, although it cannot be excluded that inhibitor treatment directly affected lymphatic vessels development, these observations suggest a requirement of *pdgfrb*+ MC for lymphatic development.

As both mutant and AG1296-treated larvae retained a substantial proportion of their MCs, we decided to eliminate *pdgfrb*+ MCs utilizing MC-selective nitroreductases (NTR) and metorodinazole (MTZ) ablation system (NTR-MTZ ablation system) (*Curado et al., 2008*), *TgBAC(pdgfrb:Gal4FF);Tg(14xUAS:3xFLAG-NTR, NLS-mCherry)*, to confirm the involvement of MCs in lymphatic vessel formation. In this transgenic line, MTZ is converted to its cytotoxic form by NTR expressed in *pdgfrb*+ MCs, which leads to selective MC death (*Figure 2—figure supplement 1A*). When ablating MCs just prior to LEC migration out from HM region by utilizing this MC-selective NTR-MTZ ablation system, LEC migration along aISV and subsequently TD formation were severely compromised (*Figure 2A–C*, *Figure 2—figure supplement 1B*). To further determine if MCs are necessary for LEC migration, we ablated MCs locally by two-photon laser just after LEC migrated out of the HM region (*Figure 2D–G*). To ensure that we did not damage the aISV during the ablations, we recorded the transmitted light videos and observed unperturbed blood flow in the aISV before and after ablation, suggesting that the vessel remained undamaged and intact (*Figure 2—figure supplement 2A*, *Figure 2—videos 3; 4*). As a control we targeted the tissue adjacent to the MCs in the same embryo (*Figure 2*). The time-lapse imaging over 5 hours post ablation (hpa) confirmed that LEC migration was dramatically inhibited in the MC-ablated group compared to the control non-MC-ablated group (*Figure 2F and H*, *Figure 2—video 1*, *Figure 2—video 2*). We also imaged the same embryos 1 day later and observed that 40%

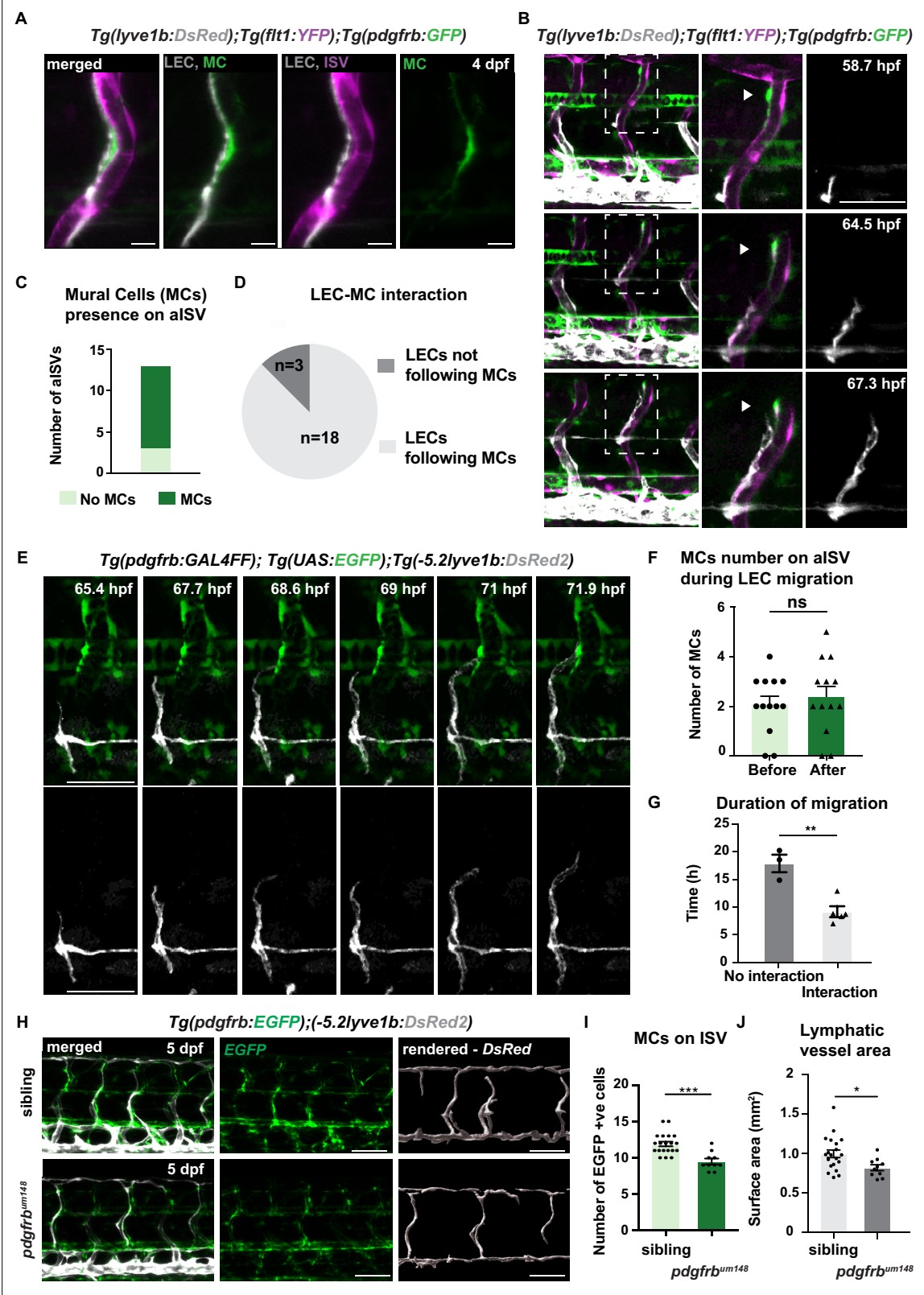

**Figure 1.** *pdgfrb*[high] mural cells (MCs) emerge around arterial intersegmental vessels (aISVs) prior to lymphatic endothelial cell (LEC) migration and provide guidance. (A) Confocal stack image of trunk aISV in 4 dpf *Tg(flt1:YFP); Tg(-5.2lyve1b:DsRed2);TgBAC(pdgfrb:GFP)* of lymphatic endothelial cells (grey, LEC), arterial intersegments vessels (magenta, aISV) and mural cells (green, MC). Scale bar; 10 μm. (B) Confocal stack images from time-lapse images in the trunk of 2 dpf *Tg(flt1:YFP); Tg(-5.2lyve1b:DsRed2); TgBAC(pdgfrb:GFP)* embryos (LECs in grey). Boxed regions are enlarged

*Figure 1 continued on next page*

*Figure 1 continued*

(right panels). Arrowheads indicate pdgfrb+ MCs (green) next to aISVs (magenta during LEC migration). Scale bars; 100 µm or 50 µm (enlarged image). (C) Quantification of aISVs with (n=10) or without (n=3) MCs presence from n=7 embryos when LECs left HM for time lapse videos as in (E). (D) Quantification of LEC and MC interaction during migration (n=10 embryos, with four somites counted per embryo). Migrating following MC n=18, migrating not following MC n=3 from time lapse videos in (E). (E) Confocal stack images from time lapse movies of LEC migration. *TgBAC(pdgfrb:GAL4FF);(UAS:GFP)* in green and *Tg(-5.2lyve1b:DsRed2)* in grey. Scale bar: 50 µm (F) Quantification of MC number around aISVs (n=14) from n=7 embryos at the start and end of the migration, quantified from time lapse videos in (E). Data are presented as mean ± SEM, unpaired two-tailed Student's t-test was used. Ns, no significance. (G) Quantification of duration of LEC migration with (n=5) or without (n=3) contacting MCs from n=6 embryos. Data are presented as mean ± SEM. unpaired two-tailed Student's t-test was used. **p<0.005 (H) Confocal stack images of *Tg(pdgfrb:GAL4FF); Tg(UAS:GFP)* (green) and *Tg(-5.2lyve1b:DsRed2)* (grey) in the trunk of sibling (top) and *pdgfrb^um148* mutant (bottom) embryos at 5 dpf. Lymphatic vessle are rendered using *lyve1b:DsRed2* channel in IMARIS s structure is rendered with lyve1b:DsRed2 channel in IMARIS (right panel). Scale bar: 100 µm. (I) Quantification of pdgfrb+ mural cell numbers around ISVs in siblings (n=20) and *pdgfrb^um148* mutants (n=10). Data are presented as mean ± SEM, Mann Whitney test was used. ***p<0.0001. (J) Quantification of surface area of lymphatic vasculature in siblings (n=20) and *pdgfrb^um148* mutants (n=10). Data are presented as mean ± SEM, unpaired two-tailed Student's t-test was used. *p<0.05.

The online version of this article includes the following video, source data, and figure supplement(s) for figure 1:

**Source data 1.** Mural cells presence on arterial intersegmental vessel (aISV) at the start of lymphatic endothelial cell (LEC) migration.

**Source data 2.** Lymphatic endothelial cell (LEC)-mural cell (MC) interaction during LEC migration.

**Source data 3.** Number of mural cells (MCs) on arterial intersegmental vessel (aISV) during lymphatic endothelial cell (LEC) migration.

**Source data 4.** Duration of migration with or without interaction with mural cells (MCs).

**Source data 5.** Number of *pdgfrb^+* mural cells around intersegmental vessels (ISVs) in *pdgfrb^um148* mutant and siblings.

**Source data 6.** Surface area of lymphatic vessels in *pdgfrb^um148* mutant and siblings.

**Source data 7.** Mural cell (MC) relocation during lymphatic endothelial cell (LEC) migration.

**Source data 8.** Total lymphatic endothelial cell (LEC) number in control and AG1296-treated embryos.

**Figure supplement 1.** *pdgfrb^high* mural cells (MCs) emerge around arterial intersegmental vessels (aISVs) prior to lymphatic endothelial cell (LEC) migration and provide guidance.

**Figure supplement 2.** Trunk of 5 days post fertilization (dpf) control and PDGFRβ inhibitor-treated embryos.

**Figure 1—video 1.** Confocal time-lapse imaging in trunk in 2 days post fertilization *Tg(flt1:YFP); Tg(-5.2lyve1b:DsRed2); TgBAC(pdgfrb:GFP)* embryos corresponding to *Figure 1B* (embryos n = 10).
https://elifesciences.org/articles/74094/figures#fig1video1

**Figure 1—video 2.** Confocal time-lapse imaging in trunk in 2 days post fertilization *Tg(lyve1b: mCherry), Tg(kdrl:TagBFP), TgBAC(pdgfrb:GFP)* embryo corresponding to *Figure 1—figure supplement 1* (embryos n > 5 embryos ).
https://elifesciences.org/articles/74094/figures#fig1video2

**Figure 1—video 3.** Confocal time-lapse imaging of *Tg(dab2:GALFF);Tg(UAS:GFP);Tg(pdgfrb:mCherry)* in trunk from 2 days post fertilization, embryo corresponding to *Figure 1—figure supplement 1C* (embryos n > 5 embryos).
https://elifesciences.org/articles/74094/figures#fig1video3

**Figure 1—video 4.** Confocal time-lapse imaging in trunk in 2 days post fertilization *TgBAC(pdgfrb:GAL4FF);(UAS:GFP), Tg(–5.2lyve1b:DsRed2)* embryos corresponding to *Figure 1E*.
https://elifesciences.org/articles/74094/figures#fig1video4

**Figure 1—video 5.** Representative confocal time-lapse imaging of lymphatic endothelial cell migrating and interacting with mural cels (corresponding to *Figure 1G*) in trunk at 2 days post fertilization.
https://elifesciences.org/articles/74094/figures#fig1video5

**Figure 1—video 6.** Representative confocal time-lapse imaging of lymphatic endothelial cell migrating without mural cell interaction (corresponding to *Figure 1G*) in trunk at 2 days post fertilization.
https://elifesciences.org/articles/74094/figures#fig1video6

of the larvae receiving MC ablation failed to form DLLV completely, while all control larvae form DLLVs (*Figure 2G and I*). Together, these data demonstrate an important role of arterial-associated MCs for the robust LEC migration.

Arterial ISVs have been previously shown to be necessary for LEC migration (*Bussmann et al., 2010*), thus MC might be a link for a direct interaction between the two cell types or an indirect effect mediated via aISVs. Importantly, the absence of MCs does not affect arterial identity at early stages in zebrafish (*Ando et al., 2019*), arguing that the importance of MC in LEC development is not simply to regulate aEC presence or abundance. Therefore, we directly tested if aEC function is critical for MC-dependent LEC migration along aISVs. We ablated ECs in aISV after the emergence of MCs using

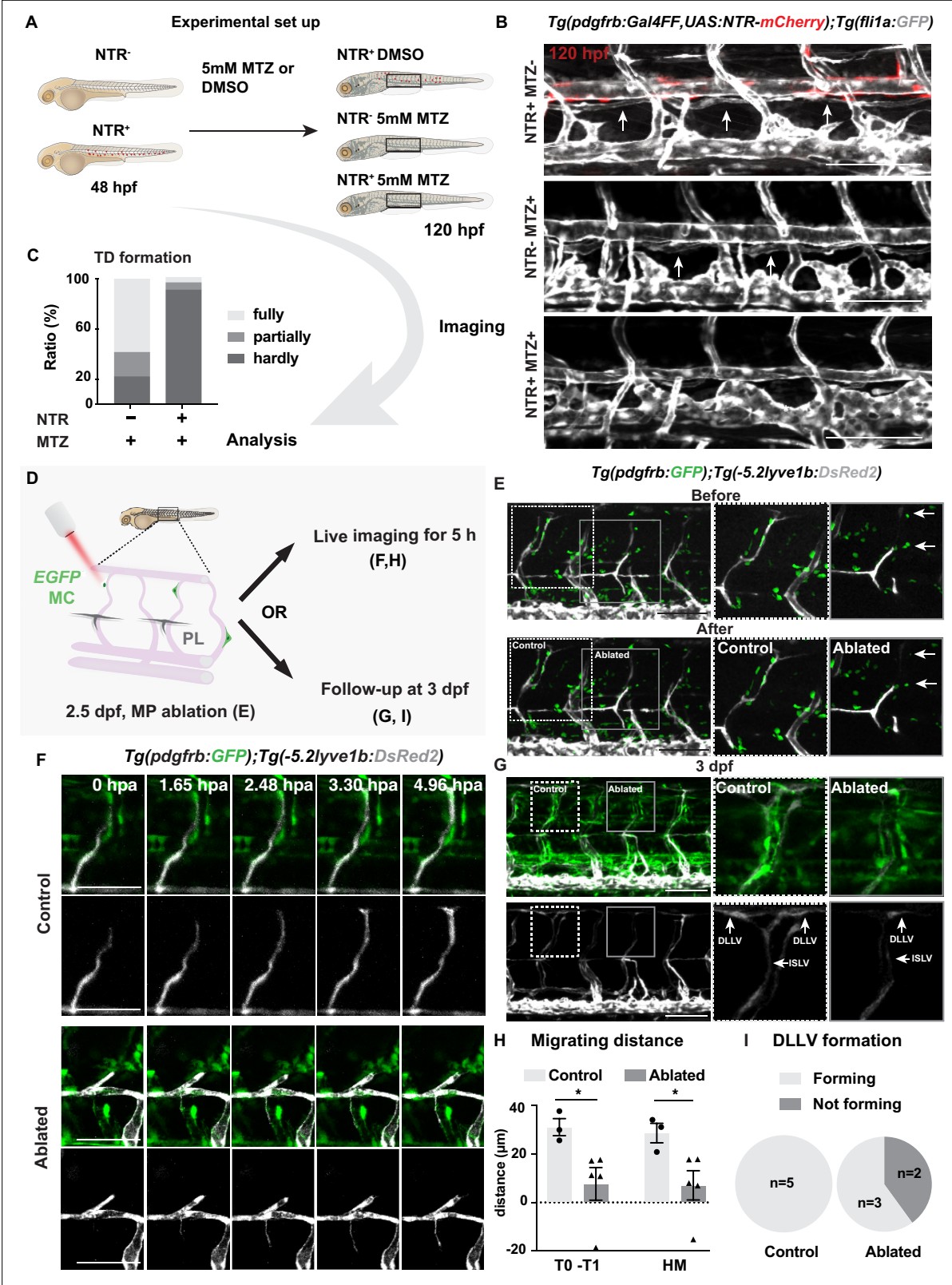

**Figure 2.** Mural cells are required for formation of lymphatic vascular bed. (**A**) Work flow of cell ablation by the nitroreductases (NTR)-metorodinazole (MTZ) system. *Tg(pdgfrb:Gal4FF);Tg(14xUAS:3xFLAG-NTR,NLS-mCherry)* (red) and *Tg(fli1a:GFP)* (grey) were imaged at 120 hours post fertilization (hpf) after treatment with DMSO or 5 mM MTZ from 48 hpf, and the formation of thoracic duct (TD) was analysed. (**B**) Confocal stack images of the trunk in 5 days post fertilization (dpf) embryos treated as described in (**A**). Arrows indicates TD forming beneath dorsal aorta. Asterisks indicate the absence

*Figure 2 continued on next page*

*Figure 2 continued*

of TD. Scale bar: 100 µm. (**C**) Quantification of (**B**). Embryos were scored as fully (completely connected TD), partially (partially formed TD) and hardly (almost or no TD visible) formed based on the TD development. In the NTR⁻ MTZ⁺ group (n = 36), n = 21 embryos with fully formed TD, n = 7 embryos with partly formed TD, n = 8 embryos with hardly formed TD were identified. In the NTR⁺ MTZ⁺ group (n = 34), n = 1 embryo with fully formed TD, n = 2 embryos with partly formed TD, n = 31 embryos with hardly formed TD were identified. Data were presented as ratio to total number of embryos analysed. (**D**) Work flow of cell ablation by multi-photon microscopy. Mural cells (MCs, green) labelled by *TgBAC(pdgfrb:GAL4FF; UAS:GFP)* and lymphatic endothelial cells (LECs) by *Tg(–5.2lyve1b:DsRed2)* (grey). MCs on intersegmental vessel in proximity to sprouting LEC were ablated at 57 hpf. For analysis, ablation was either followed by time-lapse imaging or confocal imaging at 3 dpf. (**E**) Confocal stack images before and after ablation. Control ablation (dashed box) in the adjacent region of GFP⁺ MCs and GFP⁺ MC on arterial intersegmental vessel (aISV) (solid grey box) was performed in the same embryos. Arrows indicate ablated GFP-positive cells. Scale bar: 100 µm. Middle and right panels, zoom-in images cropped in z-stacks. (**F**) Live imaging of lymphatic endothelial cell migration in the context of control (top images) and GFP⁺ MC on aISV (bottom images) after ablation, with confocal stack images from time lapse at selected timepoints from 0 to 4.96 hpa. Scale bar: 50 µm. (**G**) Confocal stack images of 3 dpf embryos in (**E**). Dashed box, control ablation. Solid grey box, MC ablation. DLLV, dorsal longitudinal lymphatic vessel; ISLV, intersegmental lymphatic vessel. Scale bar: 100 µm. (**H**) Quantification of migration distance from time-lapse videos corresponding to (**F**). Distance was calculated as both T0-T1 and the perpendicular distance between the T1 and HM for embryos with (n = 4) or without (control, n = 3) ablation. T0, the sprouting front of LECs at the start of video; T1, sprouting front of LECs at the end of video. Data are presented as mean ± SEM, unpaired two-tailed Student's t-test or Mann-Whitney test was used on two types of measurements respectively. *p < 0.05. (**I**) Quantification of DLLV formation at 3 dpf. DLLV forming (n = 3), not forming (n = 2) in the ablated group and DLLV forming (n = 5) in the control group.

The online version of this article includes the following video, source data, and figure supplement(s) for figure 2:

**Source data 1.** Thoracic duct (TD) formation at 120 hours post fertilization (hpf) presented as ratio of total.

**Source data 2.** Migration distance in control and ablated lymphatic endothelial cell (LEC).

**Source data 3.** Follow-up of dorsal longitude lymphatic vessel (DLLV) formation at 3 days post fertilization (dpf).

**Source data 4.** Migrating distance of lymphatic endothelial cell (LEC) and arterial intersegmental vessel (aISV) post arterial endothelial cell (aEC) ablation.

**Figure supplement 1.** Ablation with nitroreductases (NTR)-metorodinazole (MTZ) system.

**Figure supplement 2.** Mural cells (MCs) ablation with nitroreductases (NTR)-metorodinazole (MTZ) system.

**Figure supplement 3.** Aterial endothelial cells (aECs) ablation with multi-photon laser.

**Figure 2—video 1.** Control ablation representative confocal time-lapse imaging of two-photon cell ablation in *TgBAC(pdgfrb:GAL4FF);(UAS:GFP); Tg(–5.2lyve1b:DsRed2)*, corresponding *Figure 2F*.
https://elifesciences.org/articles/74094/figures#fig2video1

**Figure 2—video 2.** Mural cell ablation, representative confocal time-lapse imaging of two-photon mural cell ablation in *TgBAC(pdgfrb:GAL4FF);(UAS: GFP);Tg(–5.2lyve1b:DsRed2)*, corresponding *Figure 2F*.
https://elifesciences.org/articles/74094/figures#fig2video2

**Figure 2—video 3.** Representative confocal time-lapse imaging of two-photon ablation of mural cell in *TgBAC(pdgfrb:GAL4FF);(UAS:GFP);Tg(– 5.2lyve1b:DsRed2)* with transmitted light channel.
https://elifesciences.org/articles/74094/figures#fig2video3

**Figure 2—video 4.** Representative confocal time-lapse imaging of two-photon ablation of contol cell in *TgBAC(pdgfrb:GAL4FF);(UAS:GFP);Tg(– 5.2lyve1b:DsRed2)* with transmitted light channel.
https://elifesciences.org/articles/74094/figures#fig2video4

**Figure 2—video 5.** Representative confocal time-lapse imaging of two-photon ablation of arterial intersegmental vessels (aISV) in *Tg(flt1:YFP);TgBAC(p dgfrb:GFP);Tg(–5.2lyve1b:DsRed2)*, corresponding *Figure 2—figure supplement 3*.
https://elifesciences.org/articles/74094/figures#fig2video5

the two-photon laser system (*Figure 2—video 5*). We observed that even in the absence of aECs, but remaining presence of MCs, LEC migration progressed (*Figure 2—figure supplement 3*), suggesting that signals from MCs are sufficient to promote LEC migration. Thus, the previously reported strong inhibition of LEC development in aISV-depleted mutants (*Bussmann et al., 2010*) might include the effects of MC loss as aISVs ECs are essential for MC formation (*Ando et al., 2016*). However, as the aISV rapidly regrow following laser ablation system (*Figure 2—figure supplement 3B*), the long-term effects cannot be assessed, and it remains to be determined to what extend the molecular signals from MC act in synergy with aECs to promote LEC migration.

## Pdgfrb-positive pericytes express pro-lymphangiogenic factors

Our results suggest that *pdgfrb*-positive MCs play a direct role in guiding LEC migration along aISVs. To gain a better understanding of MC populations and pro-lymphatic factors that contribute to this

process, we took advantage of recently published scRNA-seq data from *TgBAC(pdgfrb:egfp)* larvae at 5 dpf (*Shih et al., 2021*). We focused on prospective MC populations by re-clustering previously identified pericyte and smooth muscle cell types (*Figure 3A*; *Figure 3—figure supplement 1*). From this analysis we found two clusters that express a previously identified pericyte gene signature (e.g. high levels of *notch3*, *pdgfrb*, and *ndufa4l2a*; *Figure 3B*; *Figure 3—figure supplement 1*), as well as pericyte-like cells that lacked the definitive pericyte marker, *ndufa4l2a* (*Shih et al., 2021*). We also noted fibroblasts marked by *pdgfra*, along with smooth muscle cell clusters expressing high levels of *desmb*, *myocd*, *cnn1b*, and *tagln*, respectively (*Figure 3B*). Additional clusters included pharyngeal arch mesenchymal cells and the cells from bulbus arteriosus (*Figure 3—figure supplement 1*).

We next assessed expression of known non-autonomous regulators of lymphatic growth, including *vegfc* and the essential Vegfc-processing factors: *ccbe1*, *adamts3*, and *adamts14* (*Bui et al., 2016*; *Hogan et al., 2009a*; *Janssen et al., 2016*; *Jeltsch et al., 2014*). We also surveyed expression of *svep1*, a putative ligand for LEC-expressed *itga9*, and the chemokines *cxcl12a* and *cxcl12b*, which have been implicated in trunk lymphatic vessel patterning (*Figure 3B*; *Cha et al., 2012*; *Karpanen et al., 2017*). We observed prominent *vegfc* expression in pericyte and pericyte-like cells, with lower levels detectable in several smooth muscle cell clusters (*Figure 3B*). However, only peri-like cells expressed *ccbe1*. Similarly, only a single SMC cluster expressed *adamts3* while *adamts14* was not detected in any clusters (*Figure 3B*). We observed *svep1* transcript at low levels in two smooth muscle cell clusters. We confirmed absence of *svep1* in MCs by analysis of *pdgfrb* and *svep1* transgenic reporters (*Figure 3—figure supplement 2A-B*). Similar to *vegfc*, *cxcl12a* was seen in multiple cell types with prominent expression in pericytes, as well as the two vegfc-expressing smooth muscle cell clusters (*Figure 3B*). By contrast, prospective trunk MC clusters were largely devoid of genes encoding functional receptors for Vegfc (*kdr*, *kdrl*, and *flt4*) or Cxcl12 (*cxcr4*). Interestingly, putative pericytes expressed *ackr3b*, the atypical receptor for Cxcl12 (*Figure 3—figure supplement 1*). To assess the expression of the chemokines receptors in LECs, we sorted double positive cells for TagRFP and nEGFP from *TgBAC(prox1a:KalTA4-4xUAS-ADV.E1b:TagRFP);Tg(fli1a:nEGFP)* embryos at 3 dpf (*Figure 3—figure supplement 3A*). We observed expression of *ackr3b* but low levels of *cxcr4a* and *cxcr4b* (*Figure 3—figure supplement 3B*), thus supporting previously reported expression of chemokine receptors in LECs (*Cha et al., 2012*).

Previous studies have identified perivascular fibroblast populations that contribute to blood or lymphatic vessel development in the zebrafish trunk (*Rajan et al., 2020*; *Wang et al., 2020*). Rajan et al. have described a *pdgfrb^low^* perivascular fibroblast that expresses *nkx3-1* and appears to be required for vessel stability. Similarly, a *pdgfra*-positive fibroblast population that expresses *vegfc*, *ccbe1*, *adamt3*, and *adamts14* can contribute to lymphatic vessel patterning in the trunk. This pro-lymphatic fibroblast population can be uniquely defined within *pdgfra*-positive fibroblasts by expression of the Engrailed paralogs, *en1a* and *-b* (*Wang et al., 2020*). To determine whether MCs identified in our current study overlapped with either of these populations, we investigated expression of *nkx3-1*, *en1a*, and *en1b* in our scRNA-seq data. Despite identification of two distinct fibroblast clusters from *pdgfrb*-positive cells, neither exhibited expression of *nkx3-1*, *en1a*, or *en1b*, suggesting that pro-lymphatic MCs identified in our study are distinct from those previously reported perivascular fibroblasts.

The *pdgfrb:egfp* cells used in Shih et al. were from whole embryos, and the pericytes noted above were therefore not necessarily associated with trunk blood vessels (*Shih et al., 2021*). Therefore, we determined expression of *vegfc*, *ccbe1*, and *cxcl12a* in (MC-enriched) EGFP-positive cells isolated from micro-dissected trunks of *TgBAC(abcc9:Gal4FF);Tg(UAS:EGFP)* larvae at 3 dpf, in comparison to negative cells (*Figure 3D–E*, *Figure 3—figure supplement 3C*; *Ando et al., 2021*). To assess the expression level of these genes in arteries, we also sorted aECs from micro-dissected trunks of *Tg(flt1;YFP)* line (*Figure 3D–E*, *Figure 3—figure supplement 3C*). We confirmed purity of MC in our sort by assessing *dll4* gene expression in both aEC and MC, where we observed clear enrichment of *dll4* in aEC (*Figure 3E*). Consistent with our scRNA-seq analysis, we found that *abcc9*-positive cells from dissociated trunks show significantly higher expression of *cxcl12a*, *vegfc*, and *ccbe1* compared to EGFP-negative cells isolated in parallel. We also detected *cxcl12a*, *cxcl12b*, and *vegfc* in arteries, but not *ccbe1* (*Figure 3E*). To confirm that *vegfc* and *ccbe1* is expressed in the MCs, we used immunostaining of BAC-transgenic lines *Tg(vegfc:Gal4; UAS:RFP; pdgfrb:GFP)* and *Tg(ccbe1:YFP;pdgfrb:Gal4; UAS:NTRmcherry)*, and observed a trend of enrichment of *vegfc* and *ccbe1* expression in the *pdgfrb^high^*

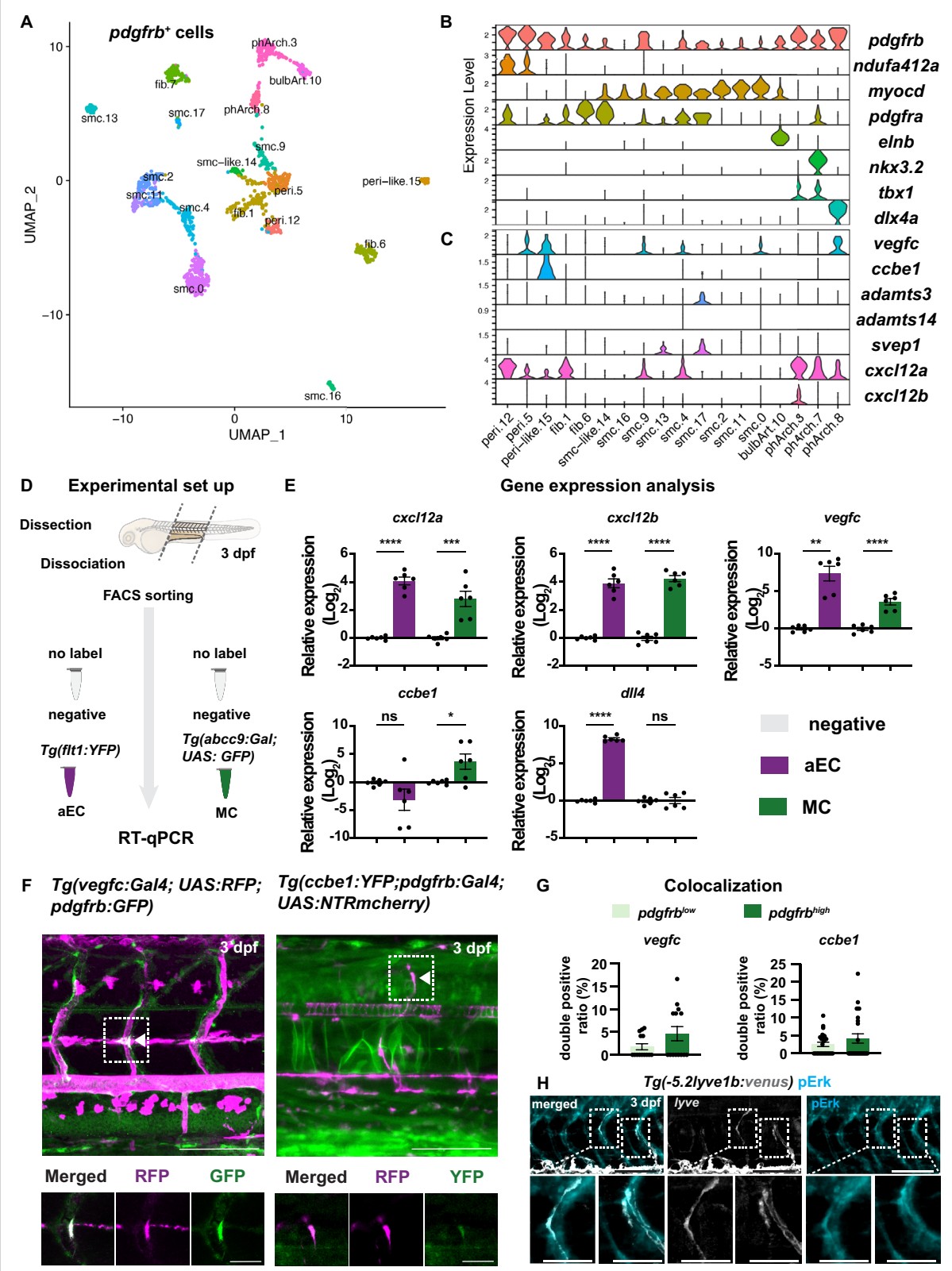

**Figure 3.** Mural cells (MCs) express chemokines and growth factors. (**A**) Uniform Manifold Approximation and Projection (UMAP) plot of smooth muscle cells and pericytes subclustered from 5 days post fertilization (dpf) *pdgfrb:egfp*-positive cells. bulbArt – bulbous arteriosus, fib – fibroblast, peri – pericyte, peri-like – pericyte-like, phArch – pharyngeal arch mesenchymal cell, smc – smooth muscle cell. (**B**) Violin plot showing markers for pericytes (*pdgfrb*, *ndufa4l2a*), smooth muscle (*myocd*), fibroblasts (*pdgfra*), bulbous arteriosus (*elnb*), and pharyngeal arch mesenchyme (*nkx3.2*, *tbx1*,

*Figure 3 continued on next page*

*Figure 3 continued*

*dlx4a*). (**C**) Violin plot showing expression of known non-autonomous pro-lymphatic factors. Expression level values are log$_2$ normalized across all cells. (**D**) Illustration of fluorescence activated cell sorting (FACS) and qPCR analysis on 3 dpf embryos. (**E**) qRT-qPCR of *cxcl12a, cxcl12b, vegfc, ccbe1,* and *dll4* in FACS sorted trunk arterial endothelial cells (aECs) and MCs cells at 3 dpf as described in (**D**). Graph represents gene expression relative to geometric average of *rpl13* and *β-actin* from three biological repeats (mean ± SEM). Unpaired two-tailed Student's t-test or Mann-Whitney test was used. No significance (ns), p ≥ 0.5. *p < 0.05, ***p < 0.0005, ****p < 0.0001. (**F**) Confocal z-projections for immunohistochemistry of fluorescent proteins in trunks of *Tg(vegfc:Gal4; UAS:RFP; pdgfrb:GFP)* and confocal image of *Tg(ccbe1:YFP;pdgfrb:Gal4; UAS:NTRmcherry)* embryos at 3 dpf. Scale bar: 100 μm; 50 μm in enlarged images. (**G**) Left panel, quantification of colocalization of *vegfc*$^+$ and *pdgfrb*$^+$ cells based on immunohistochemistry in (**F**). Right panel, quantification of colocalization of *ccbe1*$^+$ and *pdgfrb*$^+$ cells based on confocal images in (**F**), data presented as double positive ratio (mean ± SEM). Ns, no significance. (**H**) Confocal z-projections for immunohistochemistry of endogenous pERK (cyan, right) in migrating lymphatic endothelial cells (LECs) in trunks of *Tg(–5.2lyve1b:venus)* embryos (α-GFP, grey, middle) (n = 10) at 3 dpf. Scale bar: 100 μm; 50 μm in enlarged images.

The online version of this article includes the following source data, source code, and figure supplement(s) for figure 3:

**Source code 1.** SeuratCommands in R studio.

**Source data 1.** Gene expression analysis on fluorescence activated cell sorting (FACS) sorted arterial endothelial cells (aECs) and mural cells (MCs).

**Source data 2.** Colocalization of *vegfc* and *ccbe1* in *pdgfrb*$^{low}$ and *pdgfrb*$^{high}$ cells.

**Source data 3.** Colocalization of *svep1* in *pdgfrb*$^+$.

**Source data 4.** Gene expression analysis on fluorescence activated cell sorting (FACS) sorted lymphatic endothelial cells (LECs).

**Figure supplement 1.** Mural cells express chemokines and growth factors.

**Figure supplement 2.** Colocalization of *svep1* in *pdgfrb*$^+$ cells.

**Figure supplement 3.** Gating strategies of fluorescence activated cell sorting (FACS) and gene analysis of chemokine receptor on sorted lymphatic endothelial cell (LEC).

MCs on aISV compared to the rest of the *pdgfrb*$^{low}$ mesenchyme around aISV (*Figure 3F–G*). Subsequently, we determined that migrating LECs are positive for phospho(p)-ERK (*Figure 3H*), which is known to be activated downstream of Vegfc-Vegfr3 or Cxcl12-Cxcr4 signalling (*Spinosa et al., 2019*; *Xing et al., 2017*). Taken together, our molecular analysis suggests that MCs on aISVs, annotated as trunk pericytes in the transcriptomic dataset, can provide a source for essential pro-lymphangiogenic factors.

## Chemokines guide LEC migration

Chemokines have been shown to be important for LEC migration with LECs being attracted by mosaic overexpression of *cxcl12b*, whereas the *cxcl12a, cxcl12b,* and *cxcr4a* mutants show defects in TD formation (*Cha et al., 2012*). To understand if signalling mediated by these ligands is essential for LEC migration during the timepoints harmonized with MC emergence, we took advantage of temporal administration of a Cxcr4 inhibitor, AMD3100, to the embryos (*Figure 4A–B*, *Figure 4—figure supplement 1A*, *Figure 4—videos 1–4*). We added the drug at 51 hpf, after the PL had reached the HM but before continuing to move dorsally and ventrally (*Figure 4A*). Time-lapse imaging revealed that in AMD3100-treated embryos, LECs migrated shorter distances with decreased velocity compared with the controls (*Figure 4B–D*). We also investigated the filopodia formation as an indicator for proper sensing of guidance cues (*Meyen et al., 2015*), in the AMD3100-treated embryos in the *Tg(fli1a:lifeact-EGFP)* background. We found an increased number of filopodia reaching statistical significance 7.5 hours after exposure (*Figure 4E–F*, *Figure 4—videos 5–8*). We observed filopodia formation extended not only towards the migrating front but also laterally in AMD3100-treated embryos, implying the compromised directional migration of LECs. While, we did not observe the LEC apoptosis in the AMD3100-treated embryos. Together our data show that chemoattractants drive LEC migration.

## ERK activation promotes LEC migration and survival

To search for the downstream signalling of chemokines needed for proper LEC migration, we decided to assess if ERK activation is required for the LEC migration from HM, as we observed ERK activation in migrating LECs. We treated embryos with MEK inhibitor, SL327, at 51 hpf just prior to their migration and observed defects in LEC formation at 5 dpf (*Figure 4G*, *Figure 4—figure supplement 2A*). We assessed the phenotypes by time-lapse imaging (*Figure 4G–H*, *Figure 4—videos 9–12*). Tracing the migration distance revealed a reduced number of migrating cells and increased number of cells that

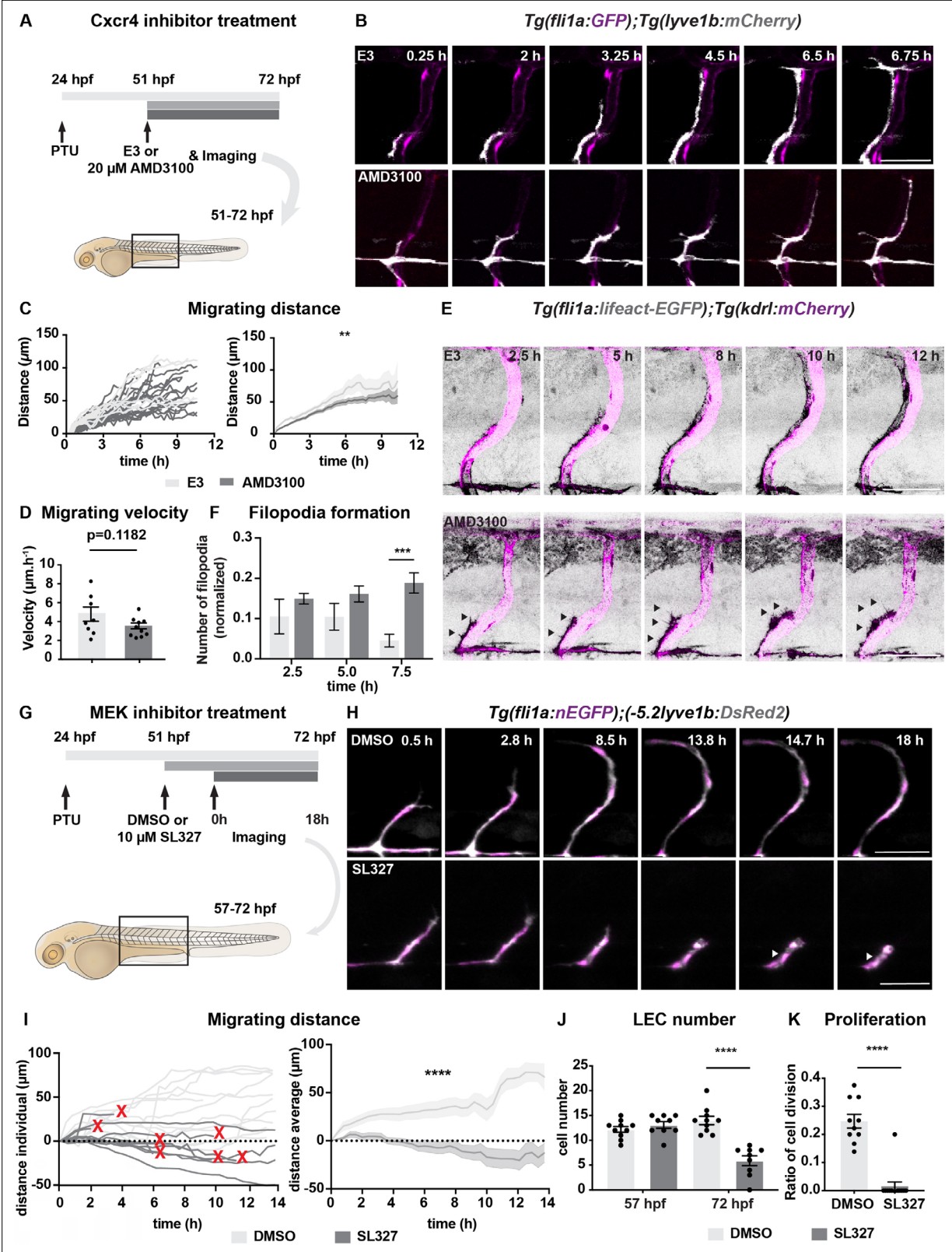

**Figure 4.** Chemokines and growth factor signalling promotes lymphatic endothelial cell (LEC) migration and survival. (**A**) Work flow of Cxcr4 inhibitor treatment. *Tg(fli1:GFP);Tg(lyve1b:mCherry)* embryos were grown in PTU (1-phenyl 2-thiourea) from 24 hours post fertilization (hpf) to prevent pigment formation, then changed to 20 µM AMD3100 or E3 water (embryo medium) at 51 hpf. (**B**) Confocal stack images from time-lapse imaging of *Tg(fli1a:GFP; lyve1b:mCherry)* embryos as indicated in (**A**). Scale bar: 50 µm. (**C**) (Left) Quantification of dorso-ventral migration showing individual tracks

*Figure 4 continued on next page*

*Figure 4 continued*

for the sprouting LECs in embryos (n = 6) in E3 water or embryos (n = 10) in AMD3100. (Right) Quantification of dorso-ventral migration showing average (mean from single tracks; left) of tracks in E3 and AMD3100-treated groups. Data are presented as mean ± SEM, unpaired two-tailed Student's t-test was used. **p < 0.005. (**D**) Quantification of velocity of dorso-ventral migration from time-lapse video described in (**B**). Sprouting front of LECs in E3 water (n = 6) and AMD3100- (n = 10) treated embryos were tracked and the distance between starting and end position of sprouting front was measured and subsequently divided by duration. Data are presented as mean ± SEM. Unpaired two-tailed Student's t-test was used. Ns, no significance, p > 0.1. (**E**) Confocal stack images from time-lapse imaging of *Tg(fli1a:lifeact-EGFP);Tg(kdrl:mCherry)* as indicated in (**A**). Arrows indicate dynamic filopodia formation during LEC migration. Scale bar: 50 μm. (**F**) Quantification of frequency of filopodia formation from time-lapse video from (**E**). Number of protrusions in LEC sprouts were counted and normalized to the sprout length, control (E3, sprouts n = 8 from 8 embryos) and treated (AMD3100, n = 12 from 10 embryos) embryos. Data are presented as mean ± SEM. Unpaired two-tailed Student's t-test was used. ***p < 0.0005. (**G**) Work flow of MEK inhibitor treatment. *Tg(fli1:GFP);Tg(lyve1b:mCherry)* embryos were grown in PTU from 24 hpf, then changed to 10 μM SL327, a MEK inhibitor, or DMSO at 51 hpf. Time-lapse imaging was started at 57 hpf. (**H**) Confocal z-stack images from time lapse of 57 hpf *Tg(fli1a:nEGFP)^{y7}* (green) and *Tg(–5.2lyve1b:DsRed2)* (grey) embryos treated with DMSO or 10 μM SL327 from 51 hpf. Grey arrowheads indicate cell death. Scale bar: 50 μm. (**I**) (Left) Quantification of dorso-ventral migration showing individual cell tracks for nuclei of sprouting LECs in DMSO- (embryos, n = 10; left panel) and SL327- (embryos, n = 9; right panel) treated embryos as described in (**H**). Red cross indicates cell death at the end of tracking. (Right) Average (mean) of tracks in DMSO- and SL327-treated groups. Data are presented as mean ± SEM, unpaired two-tailed Student's t-test was used. ****p < 0.0001. (**J**) Quantification of total LEC numbers at beginning (**T0**) and end (**T1**) of the time lapse of embryos (n = 10) in DMSO- and SL327-treated embryos (n = 9); data are presented as mean ± SEM. T0 DMSO vs. T1 SL327 p < 0.0001, T0 SL327 vs. T1 SL327 p < 0.0001, T1 DMSO vs. T1 SL327 p < 0.0001. Other comparisons were ns. One-way ANOVA with Tukey's post hoc test for statistical analysis. ****p < 0.0001. (**K**) Quantification of cell proliferation in DMSO (n = 9) and SL327-treated (n = 13) embryos as described in (**E**). Nuclear marker in green was used to count cell division events. Data are presented as mean ± SEM, Mann-Whitney test was used. ****p < 0.0001.

The online version of this article includes the following video, source data, and figure supplement(s) for figure 4:

**Source data 1.** Migrating distance quantified from control and AMD3100-treated embryos.

**Source data 2.** Migrating velocity quantified from control and AMD3100-treated embryos.

**Source data 3.** Filopodia formation quantified from control and AMD3100-treated embryos.

**Source data 4.** Migrating distance quantified from control and SL327-treated embryos.

**Source data 5.** Lymphatic endothelial cell (LEC) number before and after migration in control and SL327-treated embryos.

**Source data 6.** Lymphatic endothelial cell (LEC) proliferation in control and SL327-treated embryos.

**Source data 7.** Positive signal in TUNEL staining from control and SL327-treated embryos.

**Figure supplement 1.** Trunk of 5 days post fertilization (dpf) control and Cxcr4 inhibitor-treated embryos.

**Figure supplement 2.** Cell death in control and SL327-treated embryos by TUNEL staining.

**Figure 4—video 1.** Representative confocal time-lapse imaging of *Tg(fli1a:GFP);Tg(lyve1b:mCherry)* in E3 water zoom-in view of *Figure 4—video 2*, ,imaging from 51 hours post fertilization , embryo corresponding to *Figure 4B*.
https://elifesciences.org/articles/74094/figures#fig4video1

**Figure 4—video 2.** Representative confocal time-lapse imaging of *Tg(fli1a:GFP);Tg(lyve1b:mCherry)* in E3 water, trunk overview.
https://elifesciences.org/articles/74094/figures#fig4video2

**Figure 4—video 3.** Representative confocal time-lapse imaging of *Tg(fli1a:GFP);Tg(lyve1b:mCherry)* treated with 20 μM AMD3100 zoom-in view of *Figure 4—video 4*.
https://elifesciences.org/articles/74094/figures#fig4video3

**Figure 4—video 4.** Representative confocal time-lapse imaging of *Tg(fli1a:GFP);Tg(lyve1b:mCherry)* treated with 20 μM AMD3100, trunk overview.
https://elifesciences.org/articles/74094/figures#fig4video4

**Figure 4—video 5.** Representative confocal time-lapse imaging of *Tg(fli1a:lifeact-EGFP);Tg(kdrl:mCherry)* in E3 water zoom-in view of *Figure 4—video 6* embryo corresponding to *Figure 4E*.
https://elifesciences.org/articles/74094/figures#fig4video5

**Figure 4—video 6.** Representative confocal time-lapse imaging of *Tg(fli1a:lifeact-EGFP);Tg(kdrl:mCherry)* in E3 water, trunk overview, embryo corresponding to *Figure 4E*.
https://elifesciences.org/articles/74094/figures#fig4video6

**Figure 4—video 7.** Representative confocal time-lapse imaging of *Tg(fli1a:lifeact-EGFP);Tg(kdrl:mCherry)* treated with 20 μM AMD3100 zoom-in view of *Figure 4—video 8*, embryo corresponding to *Figure 4E*.
https://elifesciences.org/articles/74094/figures#fig4video7

**Figure 4—video 8.** Representative confocal time-lapse imaging of *Tg(fli1a:lifeact-EGFP);Tg(kdrl:mCherry)* treated with 20 μM AMD3100 trunk overview, embryo corresponding to *Figure 4E*.
https://elifesciences.org/articles/74094/figures#fig4video8

**Figure 4—video 9.** Representative confocal time-lapse imaging of *Tg(fli1a:nEGFP);Tg(–5.2lyve1b:DsRed2)* treated with DMSO, zoom-in of *Figure 4—video 10* . Imaging from 51 hourr post fertilization. Embryo corresponding to *Figure 4H*.

*Figure 4 continued on next page*

*Figure 4 continued*

https://elifesciences.org/articles/74094/figures#fig4video9

**Figure 4—video 10.** Representative confocal time-lapse imaging of *Tg(fli1a:nEGFP);Tg(–5.2lyve1b:DsRed2)* treated with DMSO, trunk overview.
https://elifesciences.org/articles/74094/figures#fig4video10

**Figure 4—video 11.** Representative confocal time-lapse imaging of *Tg(fli1a:nEGFP);Tg(–5.2lyve1b:DsRed2)* treated with 10 µM SL327, (zoom-in of *Figure 4—video 12*).
https://elifesciences.org/articles/74094/figures#fig4video11

**Figure 4—video 12.** Representative confocal time-lapse imaging of *Tg(fli1a:nEGFP);Tg(–5.2lyve1b:DsRed2)* treated with 10 µM SL327, trunk overview.
https://elifesciences.org/articles/74094/figures#fig4video12

stalled or regressed their migration in SL327-treated embryos (n = 5) (*Figure 4H–I*). In addition, we found a dramatic decrease of LEC division from 25% in controls to 1.5% in SL327-treated embryos (*Figure 4J–K*), which is in agreement with the known necessary role of Vegfc-Vegfr3 in cell proliferation (*Cao et al., 1998*). SL327 treatment induced cell death in 7 out of 12 cells, which was further confirmed by TUNEL staining (*Figure 4—figure supplement 2B-C*), suggesting that during this lymphatic developmental window ERK activation acts as a LEC pro-survival factor. Together, these results indicate that ERK activation plays vital roles in lymphatic vessel formation. While, effects of SL327 were much greater than that of AMD3100, which may imply that other pathway such as Vegfc-Vegfr3 functions together in addition to Cxcl12-Cxcr4 signalling. It has been reported that ERK activation is primarily induced by Vegfc during the LEC specification and sprouting from the PCV (*Karkkainen et al., 2004*; *Koltowska et al., 2015a*), our results showed that Vegfc-Vegfr3 signalling also instructs LEC in the subsequent migratory events from HM to establish the lymphatic vessel network in the trunk.

## Chemokine and growth factor signalling together coordinate LEC migration

As it is speculated that both Vegfc-Vegfr3 and Cxcl12-Cxcr4 signalling to be necessary for LEC migration, albeit with nuances in cellular outputs in response to these signalling cascades, we decided to investigate the interaction between these molecular entities. By utilizing the temporally controlled Vegfc trapping by expression of soluble Flt4 (Vegfr3) in conjunction with AMD3100 chemical inhibition, we first assessed the migration phenotypes (*Figure 5A*). To assess the robustness of LEC migration in chemical-treated and heat-shocked embryos, we started the treatment at 60 hpf and time-lapse imaging from 62 hpf (*Figure 5—videos 1–3*). Embryos treated with soluble Flt4 showed severe impairments in LEC migration (*Figure 5B–D*, *Figure 5—figure supplement 1A*). In addition, we observed no difference in LEC survival between in heatshock only and the combinatory-treated group (*Figure 5—figure supplement 1B*). The combinatory treatment further decreased the LEC migrating distance, but only marginally. Interestingly we observed no additive effects on the velocity of migrating LEC by the combinatory treatment, suggesting cooperation rather than mechanistic interaction of these two signalling pathways in the coordination of LEC migration. ERK activation has been shown to be induced by Vegfc-Vegfr3 but also by Cxcl12-Cxcr4 signalling (*Kukreja et al., 2005*; *Shin et al., 2016*). To assess if the cooperation of these two singling cascades is required for activation of ERK during LEC migration, we assessed ERK activation in embryos treated with soluble Flt4 or AMD3100 or a combination of both treatments. As expected, we observed a dramatic reduction in LEC numbers and ERK activation upon soluble Flt4 treatment or combinatory treatment but not in AMD3100-treated group (*Figure 5E–G*). Surprisingly, we overserved no change in ERK activation in AMD3100 treatment alone (*Figure 5E*, *Figure 5—figure supplement 2A*). Thus, these data indicate that ERK activation may be induced mainly via Vegfc-Vegfr3 signalling rather than Cxcl12-Cxcr4 signalling during LEC migration. Together, our data provide evidence that these signalling pathways cooperate to promote proper LEC migration (*Figure 5H*).

## Discussion

Our study identified a new cellular source for essential signals to promote LEC migration. The formation of lymphatic vessels in a developing embryo is a complex process where multiple tissues and cell types interact and influence each other to form a perfectly shaped and sized functional tissue.

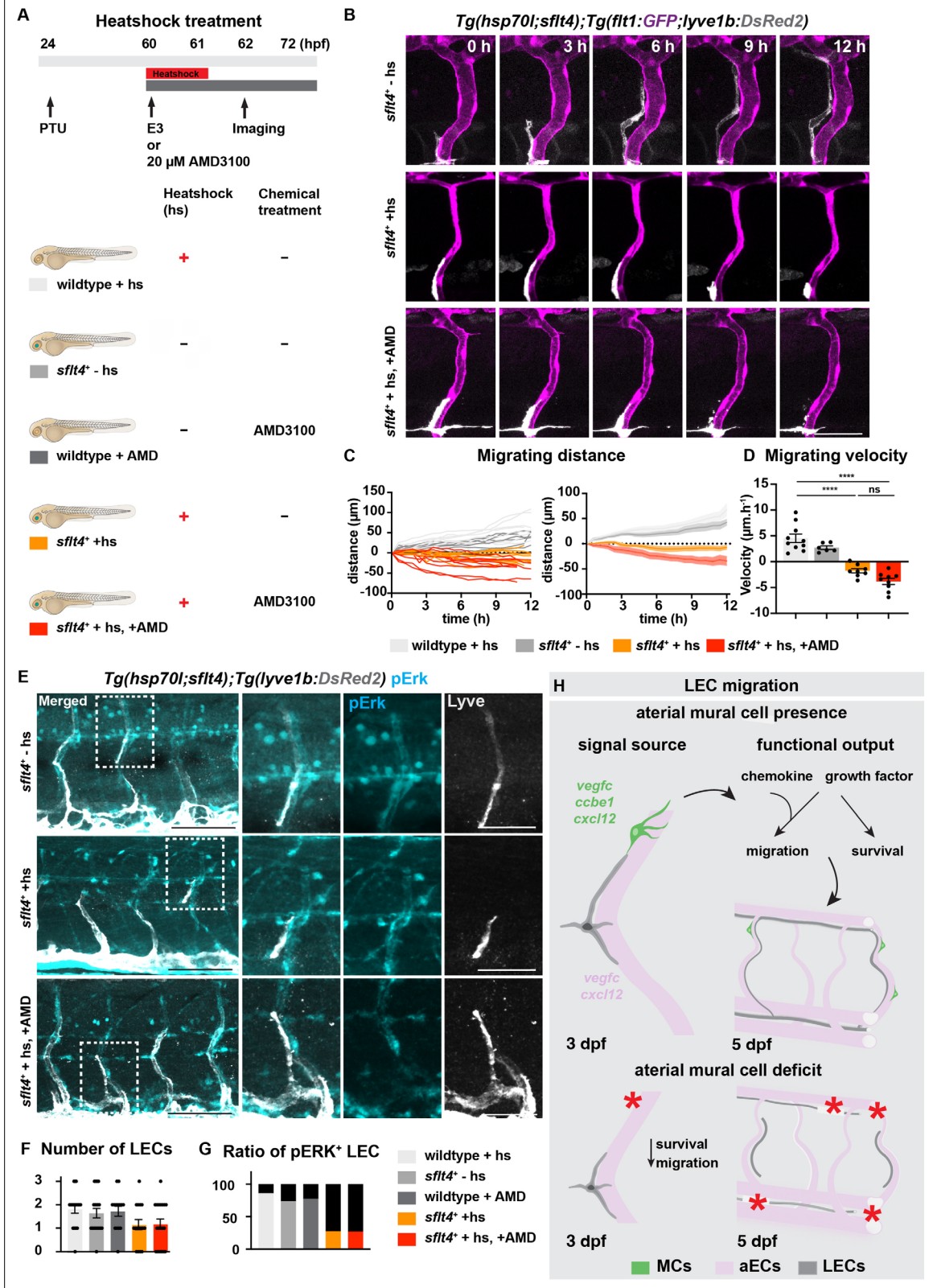

**Figure 5.** Vegfc-Vegfr together with Cxcl12-Cxcr4 coordinate lymphatic endothelial cell (LEC) migration.

The online version of this article includes the following video, source data, and figure supplement(s) for figure 5:

**Source data 1.** Migrating distance quantified from embryos without (-hs), or with heatshock (+hs), and with both heatshock and 20 µM AMD3100 (+hs,

*Figure 5 continued on next page*

*Figure 5 continued*

+AMD).

**Source data 2.** Migrating velocity quantified from embryos without (-hs), or with heatshock (+hs), and with both heatshock and 20 µM AMD3100 (+hs, +AMD).

**Source data 3.** Quantification of lymphatic endothelial cell (LEC) number in wildtype embryos with heatshock (+hs), *sflt4*[+] without heatshock (-hs), wildtype embryos treated with AMD (+AMD), *sflt4*[+] with heatshock (+hs) and *sflt4*[+] with heatshock and AMD treatment (+hs, +AMD).

**Source data 4.** Ratio of pERK[+] lymphatic endothelial cell (LEC) quantified in wildtype embryos with heatshock (+hs), *sflt4*[+] without heatshock (-hs), wildtype embryos treated with AMD (+AMD), *sflt4*[+] with heatshock (+hs) and *sflt4*[+] with heatshock and AMD treatment (+hs, +AMD).

**Source data 5.** Lymphatic endothelial cell (LEC) fate quantified from embryos without (-hs), or with heatshock (+hs), and with both heatshock and 20 µM AMD3100 (+hs, +AMD).

**Figure supplement 1.** Additional confocal image and quantification of lymphatic endothelial cell (LEC) survival related to *Figure 5A–B*.

**Figure supplement 2.** pERK in 3 days post fertilization (dpf) wildtype embryos with heatshock (+hs) and wildtype embryos treated with AMD (+AMD).

**Figure 5—video 1.** Representative confocal time-lapse imaging of *Tg(hsp70l:flt4,cryaa:Cerulean); Tg(flt1:GFP;lyve1b:DsRed2)*, from 60 hours post fertilization.

https://elifesciences.org/articles/74094/figures#fig5video1

**Figure 5—video 2.** Representative confocal time-lapse imaging of *Tg(hsp70l:flt4,cryaa:Cerulean); Tg(flt1:GFP;lyve1b:DsRed2)* from 60 hours post fertilization.

https://elifesciences.org/articles/74094/figures#fig5video2

**Figure 5—video 3.** Representative confocal time-lapse imaging of *Tg(hsp70l:flt4,cryaa:Cerulean); Tg(flt1:GFP;lyve1b:DsRed2)* from 60 hours post fertilization.

https://elifesciences.org/articles/74094/figures#fig5video3

Yet, in zebrafish, only a few tissues have been identified to supply LECs with molecular cues and promote LEC migration, among them arteries, fibroblasts, and neurons (*Bussmann et al., 2010*; *Cha et al., 2012*; *Wang et al., 2020*). With the development of transgenic lines illuminating the cellular complexity during lymphangiogenesis, new cell types become interesting targets to study the tissue-tissue interactions and instructive signals that guide LECs to progress through the embryo to their final destination. Here, we found that MCs are the source of Cxcl12, Vegfc, Ccbe1, which are necessary for Cxcl12-Cxcr4 and the Vegfc-Vegfr3 signalling pathways. Our cell ablation experiment has shown that MCs act as an accelerator which speeds up the LEC migration and in the absence of MCs the lymphatic network is incomplete. We speculate that MCs provided a signalling threshold for LECs robustly migrate from the HM region and progress dorsally across the trunk of the embryo. Interestingly, our data show that the majority of the LECs interact with MCs. Based on observations from Wang et al. and Karpanen et al., which uncovered cellular constituents for lymphangiogenic factors at earlier stages of LEC development (*Karpanen et al., 2017*; *Wang et al., 2020*) and our study revealing role for MC in LEC migration, thus it becomes evident that multiple cellular sources work together to establish the favourable levels of signalling for LEC migration.

Mechanistically our work has identified that chemokines and the Vegfc-Vegfr3 signalling pathways are specifically required during the migration of LECs across the embryo. The combinatorial treatments inhibiting both signalling pathways revealed minor additive effects suggesting that these pathways act collectively to ensure that proper lymphatic vessel formation proceeds. We have uncovered that ERK activation was mainly induced via Vegfc-Vegfr3 signalling rather than chemokine signalling. It is important to note that in the embryos exposed to soluble Flt4 we still observed residual expression of phospho-ERK in some cells. This may either indicate moderate efficiency of the soluble Flt4 transgenic tools or that additional growth factors or morphogens are also involved in the activation of the ERK pathway. However, this remains to be uncovered.

We have determined that chemokines are necessary for the progression of LEC migration and in their absence migrating cells show an increased number of filopodia, suggesting a loss in directionality. Whereas inhibition of Vegfc-Vegfr3 signalling lead to more severe phenotypes where disrupted LEC migration was accompanied by reduced proliferation and increased cell death. This uncovers that during the LEC migration Vegfc-Vegfr3 signalling has a dual role in guiding lymphatic and promoting their survival. It is important to note that the published migratory phenotypes in the *cxcl12a*, *cxcl12b*, and *cxcr4a* mutants are milder (*Cha et al., 2012*) to the ones observed in the SL327 drug treatment experiment or the cell ablation experiments. Thus, additional chemokine receptors might act together

to regulate LEC migration. We have observed higher levels of atypical receptor *ackr3b* (*cxcr7*) than *cxcr4* in LECs. The role of this receptor has been suggested to either act as a scavenger receptor to lower the signalling and ensure for correct directionality of lateral line migration (*Dona et al., 2013*) as well as it refining LEC migration in mice (*Klein et al., 2014*). The precise details of how the chemokine receptors come together to regulate LEC migration in zebrafish remains to be further characterized. Our work provides supporting evidence for chemokines and growth factor signalling pathways to come together and orchestrate proper LEC migration.

In summary, we demonstrate MC-LEC interaction at high spatio-temporal resolution during lymphatic development. We further uncovered an important role for artery-associated MCs in guidance of LECs, which is mediated by their secretion of chemoattractants including Cxcl12 and Vegfc (*Figure 5H*). Since other sources of Cxcl12 and Vegfc have already been demonstrated in the zebrafish embryonic trunk (*Cha et al., 2012*; *Wang et al., 2020*), we propose that colonization of the aISV by MCs may provide the signalling needed for robust LEC migration after moving away the HM region and migration along the aISVs. Our study underscores the importance of spatial and temporal control of the guidance cues and mitogens to promote and refine the migratory path and survival of LECs. Our finding of MC as a molecular source for lymphangiogenic factors should have relevance to future designs aiming at re-establishing lymphatic vessels in disease contexts.

## Materials and methods

### Zebrafish

Zebrafish were maintained in the Genome Engineering Zebrafish National Facility, Uppsala University, using standard husbandry conditions (*Aleström et al., 2020*). Animal experiments were carried out under ethical approval from the Swedish Board of Agriculture (5.2.18-7558/14). Previously published transgenic lines used were *Tg(fli1a:nEGFP)$^{y7}$*, *Tg(–5.2lyve1b:DsRed2)$^{nz101}$* (*Okuda et al., 2012*), *Tg(5xU-AS:GFP)* (*Asakawa et al., 2008*), *TgBAC(pdgfrb:Gal4FF)$^{ncv24}$*, (*Ando et al., 2016*) *TgBAC(pdgfrb:GFP)$^{ncv22}$* (*Ando et al., 2016*), *TgBAC(abcc9:GAL4FF)$^{ncv34}$* (*Ando et al., 2019*), *Tg(flt1:YFP)$^{hu4881}$* (*Hogan et al., 2009b*), *Tg(fli1a:GFP)$^{y1}$* (*Lawson and Weinstein, 2002*), *Tg(–7kdrl:DsRed2)$^{pd27}$* (*Kikuchi et al., 2011*), *pdgfrb$^{um148}$* (*Kok et al., 2015*), *Tg(kdrl:TagBFP)$^{mu293Tg}$* (*Matsuoka et al., 2016*), *Tg(fli1a:Myr-GFP)$^{ncv2Tg}$* (*Fukuhara et al., 2014*), *Tg(dab2:GFP)$^{ncv67Tg}$* (*Shin et al., 2019*), *Tg(hsp70l:flt4, cryaa:Cerulean)$^{bns82}$* (*Matsuoka et al., 2016*), *Tg(UAS:RFP)$^{nkuasrfp1a}$* (*Asakawa et al., 2008*), *Tg(vegfc:Gal4FF)$^{mu402}$* (*Wang et al., 2020*), *Tg(svep1:Gal4;UAS:RFP)$^{hu4767/hu4767}$* (*Karpanen et al., 2017*), *Tg(ccbe1:mCitrine)$^{hu6741}$* (*Wang et al., 2020*) (referred to as *Tg(ccbe1:YFP)*), *Tg(UAS-E1b:NfsB-mCherry)$^{c264}$* (*Davison et al., 2007*) (referred to as *Tg(UAS:NTR-mcherry)*). *Tg(lyve1:mCherry)$^{ncv87Tg}$* and *Tg(14xUAS:3xFLAG-NTR,NLS-mCherry)$^{ncv514Tg}$* were generated in this study.

### Genotyping

For *pdgfrb$^{um148}$* the following primers were used for PCR:

> *pdgfrb* Forward 5'- ATGCGCTAAAGGTGAATTGG- 3'
> *pdgfrb* Reverse 5'- GCGTCTGCCATAGTTGAACA- 3'

The PCR product was digested with Mbo1 restriction enzyme at 37°C for 1 hr. The digested product was run on 2% agarose gel. The cut of wildtype fragment results in two bands of 200 and 300 bps long; while the fragment from *pdgfrb$^{um148}$* mutants is not cut, resulting in 500 bps band; fragments from a heterozygous *pdgfrb$^{um148}$* is a combination of three fragments with bands sizes of 200, 300, and 500 bps.

### Immunohistochemistry

Immunohistochemistry was performed according to a previously published protocol (*Le Guen et al., 2014*; *Shin et al., 2016*) with the following modifications. After acetone treatment embryos were treated with Proteinase K at 10 mg/ml diluted in PBST for 35 min. Antibodies used were chicken α-GFP (1:400, ab13970 Abcam), rabbit α-DsRed (1:400, Living colors, 632,496 Takara Bio), rabbit α-Phospho-p44/42 MAPK (1:250, #4370 Cell Signaling Technology), and α-rabbit IgG-HRP (1:1000, #7074 Cell Signaling Technology). TUNEL staining was performed with In Situ Cell Death Detection Kit, Fluorescein (Merck, 11684795910) with the instruction provided by the manufacturer.

## Image acquisition

Embryos were anaesthetized and mounted in 1% low-melting agarose on a 35-mm diameter glass-base dish (627870 or 627861 Greiner). Confocal images were obtained using a Leica TCS SP8 confocal microscope (Leica Microsystems) equipped with water immersion 25× (Fluotar VISR, 0.95 NA) objective, water immersion 40× (HC PL APO CS2, 1.1 NA) objective and glycerol immersion 63× (HC PL APO CS2, 1.3 NA) objective or FluoView FV1000/FV1200/FV3000 confocal upright microscope (Olympus) equipped with a water immersion 20× (XLUMPlanFL, 1.0 NA) lens. The 473 nm (for GFP), 559 nm (for mCherry), and 633 nm (for Qdot 655) laser lines in FluoView FV1000/FV1200/FV3000 confocal microscope and the 488 nm (for GFP) and 587 nm (for mCherry) in Leica TCS SP8 confocal microscope were employed, and 488 and 651 nm on the Zeiss NLO710, respectively.

## Image analysis

Image quantification was performed using z-stacks in ImageJ 2.0.0 (*Schindelin et al., 2012*), Olympus Fluoview (FV10-ASW, FV31S-SW), or IMARISx64 9.5.1 software (Bitplane). Total LEC number was counted manually using the overlay of DsRed and GFP channels over five somites in the trunk. Lymphatic vessel area was calculated by rendering the surface using DsRed channel, the non-lymphatic structures were manually removed. Measurements of surface area were exported directly from Imaris (Bitplane).

## Cell Tracking

To quantify the migrating distance, the centre of PL nuclei in *Figure 4H* was manually tracked until either cell died or disappeared from the view in Imaris (Bitplane). The individual cell track was generated by 'spot' and 'cell track' function and then manually edited if needed. The data were exported to GraphPad for plotting and statistical analysis.

## Distance measurement

The migrating distance in all figures was measured in three dimensions using the spot function in Imaris (Bitplane), see cell tracking. In *Figure 3B–C*, the sprouting front of PL just migrating away from the HM region was chosen as start point (T0) and the migrating front at the end of the time-lapse video was chosen as the end point (T1), respectively. A direct line was used to connect the dots and the length of the line segment was measured. In *Figure 2H*, the perpendicular distance between point T1 and HM was also measured in addition to the measurement above.

## Chemical treatment

To inhibit Cxcr4 signalling, the embryos were treated in 20 µM antagonist AMD3100 (Merk) diluted in E3 water (embryo medium) (*Westerfield, 1993*) from 51 to 72 hpf. To block phosphorylation and activation of ERK1/2, embryos were treated in 10 µM SL327 (EMD Millipore) diluted in E3 water with 1% DMSO, from 51 to 72 hpf. Embryos were anaesthetized and mounted in 1% low-melting agarose in a two-well slide with separate chambers, which allows spontaneous imaging of both groups. The prepared chemical solution (3 ml) was added on top of the agarose layer in one chamber and control medium (E3 water or 1% DMSO in E3 water) (3 ml) to the other chamber.

## FACS and qPCR analysis

Embryos of *Tg(flt1:YFP)* and *Tg(abcc9:Gal; UAS:GFP)* were collected at 3 dpf and screened as described in *Figure 3D*, dissociation was performed as previously described (*Kartopawiro et al., 2014*). The dissociated cells were sorted using a fluorescence activated cell sorting (FACS) Aria III (BD Biosciences) into 300 µl TRIzol LS Reagent (Thermo Fisher). Total RNA was extracted using the Quick-RNA Microprep kit (Cambridge Bioscience) following the manufacturer's instructions. RNA quality and concentration were determined using 2100 Bioanalyser Instrument (Agilent) together with Bioanalyzer High Sensitivity RNA Analysis Kit (Agilent). One ng of RNA template was subjected to cDNA synthesis using SuperScript VILO cDNA Synthesis Kit (Thermo Fisher). The synthesized cDNAs were amplified in parallel using SsoAdvanced PreAmp Supermix (Biorad), and both of the amplified samples were included for further analysis. The qPCR analysis was performed using the primers in Table S1 on CFX384 Touch Real-Time PCR Detection System (BioRad). Data were analysed using the CFX Maestro Software (BioRad). The geometric average of *rpl13* and *β-actin* or *kdrl* and *β-actin*

expression was used as a reference to calculate relative gene expression of target genes with the ddCT method and the values were presented as log or normalized log fold. Primer sequences listed in the below table.

| Target gene | Forward primer sequence 5'–3' | Forward primer sequence 5'–3' | Reference |
|---|---|---|---|
| *β-actin* | CGAGCTGTCTTCCCATCCA | TCACCAACGTAGCTGTCTTT | Designed for this study |
| *rpl13* | CATCTCTGTTGACTCACGTCG | CATCTTGAGCTCCTCCTCAGTAC | Designed for this study |
| *cxcr4a* | CATGACAGACAAGTACCGTCT | TGCTGTACAAGTTTACCGTGTA | qPrimerDB |
| *cxcr4b* | TGCTAACATTCCTGATAAGACC | GTACTTTTATTGCCAGACCTAAAGG | qPrimerDB |
| *cxcl12a* | GCAAGTGCTTTGACACAAAAG | TTTGTTTGGCAAAGTAACCCTG | qPrimerDB |
| *cxcl12b* | GATCGTGATAGCTTTGTGAACC | AATGTTAACAATGCTTGGCCTC | qPrimerDB |
| *vegfc* | TCTTAAAAGGGAGACGGTTTCA | TACATTTCCTTCTCTTGGGGTC | qPrimerDB |
| *ccbe1* | AGTGTCTGAAATGATCTACCCG | ACTTCTCTGTCTACATCCTCCT | qPrimerDB |
| *dll4* | GGACAAATGCACCAGTATGC | GTTTGCGCAGTCGTTAATGT | *Ando et al., 2019* |
| *ackr3b* | TGAACTTCTCAACTCTTGACGA | TACAGGTGAGTCTCATAACGTG | qPrimerDB |

## Ablation with multi-photon microscopy

For MCs ablation, embryos of *Tg(–5.2lyve1b:DsRed2);Tg(pdgfrb:GFP)* were laterally mounted in 1% low-melting point agarose at 57 hpf. An aISV with migrating LEC was chosen randomly, a GFP-positive MC was identified using 488 nm laser. MCs located ahead of migrating route were ablated using a two-photon laser at 790 nm (Mai Tai, Spectr-Physics Millenia PRO). Control ablations were performed as above but the adjacent area to the pdgfrb⁺ cell targeted with the two-photon laser. For aISV ablation, *Tg(–5.2lyve1b:DsRed2);Tg(flt:YFP);Tg(pdgfrb:GFP)* embryos were prepared as described above and an aISV, with LEC migrating along, was ablated using the two-photon laser targeting the connection point of dorsal longitudinal anastomotic vessel and aISV as well as proximal end of the aISV at the connection point to the DA. Larvae were imaged before and after ablation with a Zeiss LSM 710 FCS confocal microscope, which was followed by either time-lapse imaging for around 5 hours or follow-up confocal imaging at 3 dpf.

## Ablation with NTR-MTZ system

Embryos from *TgBAC(pdgfrb:Gal4FF);Tg(UAS:NTR, mCherry);Tg(fli1a:GFP)* were collected and screened as described in *Figure 2A*. The embryos were treated with E3 medium containing either 5 mM MTZ or DMSO from 48 hpf and the media was replaced daily with fresh ones. The TD was imaged at 120 hpf above the yolk extension spanning across eight to nine somites and the quantification of TD was performed by categorizing the extend of formed TD into three groups: fully (completely connected TD), partially (partially formed TD), and hardly (almost or no TD visible). The category percentages for all embryos per treatment group were calculated.

## Dye injections

Qtracker 705 Vascular Labels (ThermoFisher) diluted with (1:1) was injected into the common cardinal vein using a small capillary needle. Circulating Qtraceker 705 fluorescent dye in the blood vessels was visualized soon after the injection by confocal microscopy. Over time, in addition to blood vessels, lymphatic vessels were labelled by Qtracker 705 fluorescent dye.

## scRNA-seq analysis to assess MC expression of pro-lymphatic factors

To assess expression of known pro-lymphatic genes in MCs, we re-analysed previously published scRNA-seq data from TgBAC(*pdgfrb:egfp*)ⁿᶜᵛ²² larvae at 5 dpf. Originally processed Cell Ranger outputs from this dataset are available at GEO (GSE176129). To identify cells for re-analysis, we used the RData object from previous clustering (*Shih et al., 2021*). All of the following analysis was performed in RStudio running R4.0.5 and using Seurat 4.0.3 (*Hao et al., 2021*). The following is an

overview and we refer readers to *Figure 3—source code 1* for accompanying annotated commands used for this analysis. We first manually selected clusters expected to contain prospective MC populations. These were named as follows in the original clustering: '39-pericyte','14-smc', '52-smc', '17-smc', '51-smc', '53-smc', '6-smc'. We subsequently generated a list of barcodes comprising cells in these clusters and used those to obtain the original raw count data for these cells. We then performed normalization, identification of variable features, scaling, and clustering as described previously (*Shih et al., 2021*; *Figure 3—figure supplement 1A-B*). Previously identified genes for pericytes, smooth muscle cells, and fibroblasts were initially used to assign cell cluster identities. To classify remaining clusters (pharyngeal arch, bulbus arteriosus), we identified all cluster-specific markers and used these to find remaining lineage-defining genes.

### Heatshock treatment

Embryos of *Tg(hsp70l:flt4, cryaa:Cerulean)*[bns82]; *Tg(flt1:GFP; lyve1b:DsRed2)* were raised at 28.5°C and then subjected to 37°C heatshock for 1 hour by replacing the E3 water plus PTU with fresh pre-warmed (37°C) one and then kept in a 37°C incubator. For the non-heatshock-treated group, embryos were kept at 28.5°C for the whole time (*Figure 5A*).

### Statistical analysis

Statistical analysis was performed using Prism software (GraphPad). Gaussian distribution of samples was tested with Shapiro-Wilk normality test. Student's t-test was used for comparison of two means. For not normal distributed data, Mann-Whitney test was used for comparison of two means. One-way ANOVA with post hoc test was used for multiple comparison as stated in corresponding figure legend.

## Acknowledgements

This work was supported by Wallenberg Academy Fellowship (2017.0144), Ragnar Söderbergs Fellowship (M13/17), Vetenskapsådet (VR-MH-2016–01437), and Jeanssons Foundation. MH and SS-M were supported by funds from the DFG (CRC1348B08). The SciLifeLab Zebrafish facility in Uppsala hosted zebrafish. FACS was performed at BioVis at Uppsala University. We are grateful to A Chiba, H Nakajima, S Yuge, T Babazono, W Koeda, K Hiratomi, M Sone, E Nakamura, K Kato, and H Ichimiya for technical assistance.

## Additional information

### Funding

| Funder | Grant reference number | Author |
| --- | --- | --- |
| Knut och Alice Wallenbergs Stiftelse | 2017.0144 | Katarzyna Koltowska |
| Ragnar Söderbergs stiftelse | M13/17 | Di Peng |
| Vetenskapsrådet | VR-MH-2016-01437 | Katarzyna Koltowska |
| Jeanssons Stiftelser | | Katarzyna Koltowska |
| Deutsche Forschungsgemeinschaft | CRC1348B08 | Melina Hußmann Stefan Schulte-Merker |
| National Institutes of Health | R35HL140017 | Nathan D Lawson |

The funders had no role in study design, data collection and interpretation, or the decision to submit the work for publication.

### Author contributions

Di Peng, Koji Ando, Conceptualization, Formal analysis, Writing – original draft, Writing – review and editing; Melina Hußmann, Marleen Gloger, Formal analysis, Methodology; Renae Skoczylas,

Methodology; Naoki Mochizuki, Resources; Christer Betsholtz, Shigetomo Fukuhara, Stefan Schulte-Merker, Resources, Writing – review and editing; Nathan D Lawson, Formal analysis, Resources, Writing – review and editing; Katarzyna Koltowska, Conceptualization, Funding acquisition, Supervision, Writing – original draft, Writing – review and editing

#### Author ORCIDs
Di Peng  http://orcid.org/0000-0002-7166-730X
Koji Ando  http://orcid.org/0000-0002-4152-5706
Marleen Gloger  http://orcid.org/0000-0002-3319-7642
Renae Skoczylas  http://orcid.org/0000-0002-8570-7368
Nathan D Lawson  http://orcid.org/0000-0001-7788-9619
Katarzyna Koltowska  http://orcid.org/0000-0002-6841-8900

#### Ethics
Animal experiments were carried out under ethical approval from the Swedish Board of Agriculture (5.2.18-7558/14).

#### Decision letter and Author response
Decision letter https://doi.org/10.7554/eLife.74094.sa1
Author response https://doi.org/10.7554/eLife.74094.sa2

## Additional files

### Supplementary files
• MDAR checklist

### Data availability
All data generated or analysed during this study are included in the manuscript and all the source are uploaded.

The following previously published dataset was used:

| Author(s) | Year | Dataset title | Dataset URL | Database and Identifier |
|---|---|---|---|---|
| Shih Y-H, Portman D, Idrizi F, Grosse A, Lawson ND | 2021 | Integrated molecular analysis identifies new developmental pericyte markers in zebrafish | https://www.ncbi.nlm.nih.gov/geo/query/acc.cgi?acc=GSE176129 | NCBI Gene Expression Omnibus, GSE176129 |

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
