## [Editor Report]

Utilizing the zebrafish models and the state-of-art approaches, this study clearly identifies that the arterial mural cells serve as the source for the chemokines and growth factors, which are key factors not only for the migration and survival of lymphatic endothelial cells but also for the building of lymphatic networks in most organs during embryonic development. The findings and conclusion would be an important platform in the future design for rebuilding lymphatic vessels in treating lymphatic-deficient diseases including lymphedema.

---

## [Decision Letter]

[Editors' note: this paper was reviewed by Review Commons.]

---

## [Author Response]

Reviewer 1:– What is their relative importance and contribution (concerning provision CXCL12 and VEGFC) to the process.

Experimental plan

We thank the reviewer for this important question. To address this in more depth we will perform the following experiments:

We will utilize a genetic tool that can be temporally controlled by the heat shock promoter (hs), induce soluble *flt4* expression (*hs:sflt4* line). Soluble Flt4 is known to sequester Vegfc and prevent signalling necessary to drive lymphangiogenesis. This line has been successfully used (Harrison et al., 2019; Matsuoka et al., 2017). We will induce the *sflt4* expression at the time of lymphatic endothelial cell (LEC) migration (embryonic day 3), and assess the induction of phospho-Erk in the absence of the Vegfc-Vegfr3 signalling. In conjunction we will treat embryos with Cxcr4 inhibitor and also stain for phospho-Erk, to compare the activity levels with the *hs:sflt4* line. This will allow us to assess whether other signalling pathways also induce ERK in LECs or if Vegfc-Vegfr3 is the dominant pathways.

To assess the combinatorial requirement of Vegfc-Vegfr3 and CXCR-CXCL signalling we will simultaneously treat the embryos with Cxcr4 inhibitor and express the *sflt4*, then assess if the impairment of LEC migration is even more severe. It is important to note that the published migratory phenotypes in the *cxcl12a*, *cxcl12b* and *cxcr4a* mutants are milder to the ones observed in the SL drug treatment experiment or the cell ablation experiments. Thus, we think that these signalling pathways function together to orchestrate the LEC migration.

Conducted experiments

In accordance with our plan, we have conducted all the above experiments and found the combinatory treatment of *sflt4* and AMD3100 drug further decreased the distance LEC can migrate, but only marginally and with no further impact on the velocity of the migrating cells (new Figure 5A-D). We have found that pERK was reduced in *sflt4* treated embryos but not in AMD3100 treated ones, furthermore the combinatory treatment did not show an additional reduction in the staining. (Figure 5E-F). Based on these results we concluded that chemokines and growth factors coordinate LEC migration. Thus, these results provide new mechanistic insights of the regulation of LEC migration and activation of pERK in the migratory context.

– A deeper characterization of the MCs could involve expression analysis of e.g. of CCBE1 and SVEP1, which might provide additional insights into the underlying mechanisms of lymph vessel formation.

Experimental plan

To further understand the role of MC in the processing of the Vegfc, we will assess the expression levels of *ccbe1*, *adamts3* and *adamts14* in MCs sorted from *abcc9:GFP* line and in parallel the high and low expressing cells from *pdgfrb:GFP* line. We will also include Svep1 as it has been shown to promote LEC migration. To further support our observation in collaboration with Prof. Stefan Schulte-Merker we will perform an analysis of the coexpression of *ccbe1*, *vegfc*, *svep1* using the BAC transgenic lines out-crossed with the *pdgfrb* line.

Conducted experiments

In collaboration with Nathan Lawson, we have added single cell RNA-sequencing data of *pdgfrb* positive cells which shows that *vegfc, ccbe1 and cxcl12a* are expressed in pericytes and pericyte-like cells (Figure 3A-C, Figure3—figure supplement 1A-D). The expression of *adamts3* was found in one smooth muscle cells (SMC) cluster, *svep1* was also expressed although at low levels two SMC clusters, whereas *adamts14* was not detected in any clusters (Figure 3A-C). We have confirmed the expression of *cxcl12a*, *cxcl12b*, *vegfc* and *ccbe1* in mural cells sorted from the *abcc9:GFP* line (Figure 3 D-E). Utilizing transgenic lines we have also confirmed that *vegfc* and *ccbe1* are increased in the *pdgfrb:GFP* high expressing cells (Figure 3F-G), which is not that case for *svep1* positive cells (Figure3—figure supplement 2A-B). Together the new data strengthen the evidence that MCs are a source of prolymphangiogenic factors.

Reviewer 2:1. Characterization of the interaction between LECs – MCs.a. It will be ideal to use Flt1 for Figure 1B to distinguish between arteries and veins.

We will perform additional time-lapse imaging using the *flt1* line.

The time-lapse is added Figure 1B and Figure 1—video 1 and the text.

2. Dissociation between aECs and MCs might be difficult and there is a possibility that FACS sorted MCs have aECs contamination. It is recommended that aECs markers be examined by qRT-PCR to validate that MCs are not contaminated by aECs. There are also cxcl12a and cxcl12b reporter lines that can be used to validate their expression in MCs.

Experimental plan

This is an important issue; We will change our FACS strategy to address this.

We will sort the arterial endothelial cells from fish with *flt1:YFP* transgene only.

For MC isolation, we will use fish carrying the *abcc9:GF*P only.

For LECs and rest of EC, we will use fish with *prox1:RFP* and *nfli:GFP* (double positive – LEC, single GFP positive – all endothelial cells).

Conducted experiments

We have added data from the new cell sorting strategy. Using the *dll4* gene we have validated the purity of the MC cell population as we observed no expression of in the *abcc9:GFP* sorted cells and high expression in the *ftl1:YFP* sorted cells (Figure 3D-E). The sorting plots are shown in (Figure3—figure supplement3).

In the new sorted LECs we have observed relatively low expression of *cxcr4a* and *cxcr4b* in comparison to our initial sorts where there was a mixed LEC and VEC population. Therefore we have analysed the expression of atypical chemokine receptors a*ckr3b (cxcr7)* which have been shown to be involved in lymphatic vessel development in mice (Klein et al., 2014). We observed high expression of this receptor in the new sorted LEC. Our unpublished data from single cell RNA-seq (double positive cells sorted from *lyve:Venus;nfliCherry)* supports this observation. See Author response image 1.

**Author response image 1. sa2fig1:** Single cell RNA-seq analysis of LEC and VEC populations at 5 dpf. Top panel: clustering of cells, LECs marked by prox1a expression. Bottom: expression of chemokine ligands and receptors in LEC and VEC cell population.

3. The authors proposed that MC cells provide a signalling threshold for LECs to migrate from HMs. They might speculate why some of the LECs do not interact with MCs. Are there higher levels of expression of lymphangiogenic factors in other local tissues? This might be assessed using a combination of transgenic lines.

Experimental plan

Based on our observation and the study by Wang et al., 2020 we think that indeed that is the case. Although Wang et al. looked at fibroblast at earlier stages of LEC development, we think that there are multiple sources of Vegfc or Cxcl12 that come together to establish the favourable levels of downstream signalling for LEC migration. Utilizing the established transgenic reports for *Vegfc* and *Ccbe1*, we will address their expression levels in MCs and other tissues during the LEC migration.

Conducted experiments

In collaboration with Stefan Schulte-Merker we have analysed the *vegfc, ccbe1* and *svep1* expression using the transcriptional reporters. We observed tendency of higher percentage of *pdgfrb:GFP* high expressing cells co-expressing *vegfc, ccbe1* compare to *pdgfrb:GFP* low cells (Figure 3F-G). This was not the case *svep1* expression (Figure3—figure supplement 2A-B). This further supports our conclusions that MC are a selective source of prolymphangiogenic factors.

Reviewer 1Figure 3J and K: The bar diagrams represent statistically rather small differences (p > 0.1). Is this evidence sufficient to support the statements: "… and found LECs migrated less and their migration velocity is decreased (Figure 3G-J). This coincided with an increased number of filopodia formation on LEC (Figure 3 K),…" ?

Experimental plan

We will perform additional experiments and analyses of filopodia behaviours to better support the conclusion. Please note that the n-number were low in this experiment which impacted the significance, with extra replicate this will be resolved.

Conducted experiments

We have added new imaging and analysis of filopodia using the *fli1:lifeact-GFP* transgenic lines and observed a drastic increase in filopodia formation in the AMD3100 treated embryos (Figure 4E-F).

Suppl. Figure 3B: PDGFR-B expression in the aEC population is of concern as it might indicate a contaminated population. Could the purity of the different sorted populations be demonstrated by probing for the presence / absence of additional relevant genes?

Experimental plan

From our previous unpublished data, we saw a trend of *pdgfrb* expression in the arteries at early development stage but not in older fish. We will change our sorting strategy to sort arteries from *flt1:YFP* embryos only (without PDGFR transgene). As flt1 is not expressed in mural cells we only sort for aEC, this will resolve any potential contamination issue and will confirm/disproof the initial observation.

Conducted experiments

We have added data from the new cell sorting strategy. Using the *dll4* gene expression we have validated the purity of the MC cell population as we observed no expression in the *abcc9:GFP* sorted cells and high expression in the ftl1:YFP sorted cells (Figure 3D-E). The sorting plots and additional q-PCR analysis is presented in (Figure3—figure supplement 3AC).

Figure 4: Given the universal function of the Erk-pathway it appears problematic to attribute Erk activation mainly to VEGFR-3 activation, there might be additional growth factors or morphogens acting during the process that engage the ERK-pathway.

Experimental plan

We agree with the reviewer. We will address this experimentally by performing the phosphoERK stainning on the embryos treated with *hs:flt4* and Cxcr4 inhibitor.

Conducted experiments

We have added new experiments showing the induction of pERK downstream of Vegfc-Vegfr3 signaling but not chemokines (new Figure 5).

We have copied here the response to the comment from Reviewer 1, who had a similar question:

In accordance with our plan, we have conducted all the above experiments and found the combinatory treatment of *sflt4* and AMD3100 drug further decreased the distance LEC can migrate, but only marginally and with no further impact on the velocity of the migrating cells (new Figure 5A-D). We have found that pERK was reduced in *sflt4* treated embryos but not in AMD3100 treated ones, furthermore the combinatory treatment did not show an additional reduction in the staining. (new Figure 5E-F). Based on these results we concluded that chemokines and growth factors coordinate LEC migration. Thus, these results provide new mechanistic insights of the regulation of LEC migration and activation of pERK in the migratory context.

– In depth characterization of the identity and function of the relevant MCs (how homogeneous is the PDGFR-ß+ population)

Experimental plan

The heterogeneity of the *pdgfrb* positive cell population has been previously described (Ando *et al.*, 2019; Shih *et al.,* 2021). The cells labelled in the *pdgfrb:GFP* line can be subdivided into the *pdgfrb:GFP*^high^ and *pdgfrb:GFP*^low^ cells. Recent single cell RNA sequence (scRNAseq) analysis clearly show that *pdgfrb:GFP*^high^ cells include mural cells as well as other many cell types (Shih *et al.,* 2021). Yet, this scRNAseq data revealed that *pdgfrb:GFP*^high^ mural cells on intersegmental vessels (ISVs), that we analyzed in this paper, are seemingly a homogeneous population as they form the single cell cluster. While, *pdgfrb:GFP*^low^ cells are considered to be fibroblasts (Wang *et al.*, 2020) and precursors to mural cells (Ando et al., 2019; Rajan et al., 2020). Given this background, to address the gene expression in mural cells covering ISV where we found important for lymphatic guidance, we have sorted *abcc9* reporter positive mural cells in the restricted trunk region because *abcc9* reporter is selective in mural cells on ISV. Taking this advantage of selective gene analysis in ISV-mural cells, we found the expression of *cxcl12* and *vegfc.* Based on the published and pre-print data we think that performing additional analysis of the heterogeneity of this cell population would be redundant with what is currently available.

Conducted experiments

To address the heterogeneity of MC population labeled by *pdgfrb:EGFP*, in collaboration with Nathan Lawson we have included re-clustered data from the (Rajan et al., 2020) pre-print. We have observed 2 pericyte clusters and one pericity-like cluster (Figure 3A-C). We have also provided additional evidence of the intensity of *pdgfrb* expression in *pdgfrb:EGFP* transgenic line by generating a heat-map based on fluorescent intensity (Figure1—figure supplement 1A) which indicated that the *pdgfrb:EGFP* expressing cells are located around the intersegmental arteries.

Reviewer 2:1. Characterization of the interaction between LECs – MCs.b. Is the vessel on the right in Figure 1B also interacting with lymphatic vessels (lyve1b:mCherry)? Are there MCs along this vessel?

The vessel on the right is a vein and it is lumenised, co-expressing *lyve1b* and *kdrl,* and also connected to the posterior cardinal vein (PCV). In contrast, the LEC on the left lacks the lumen, lost connection to the PCV and expression of *kdrl*. The green cell on the vessel to the right looks like a MC. It has been shown previously that the MCs do reside along the veins (Ando *et al.*, 2019), especially their dorsal part where the arterially derived part of the vein.

c. In Supp. Figure 1, very few MCs are visible. It does not seem that the majority of LECs interact with MCs. The LECs to the left of lymphatic vessels marked by the arrowhead do seem to have MCs but the LECs are cropped out for the 60 and 69 hpf images.

The aim of this images is to show that the migration of LECs happens after MCs emergence along the intersegmental vessels. We have now provided uncropped images where more LECs can be observed in the *Tg(dab2:GALFF);Tg(UAS:GFP)* in grey and *Tg(pdgfrb:mCherry)* in green. We also added an uncropped movie from the figure 1E (*Tg(lyve1b:DsRed*) in grey and *Tg(pdgfrb:GFP*) in green). See new Supplementary Movie S2 and S3 and Supplementary Figure 1B-C.

d. Are MCs marked by arrowheads in Figure 1B also migrating? However, the MCs in Figure 1E seem to stay at the same position during the similar time frame.

We have quantified the percentage of relocating MCs, see Supplementary Figure 1B and incorporated in the text.

e. How many embryos were analyzed in Figure 1C and 1D? Similarly, are the numbers of LECs quantified in one embryo? There are no statistics.

We apologize for this oversight. We have provided the missing numbers in the figure legend.

Reviewer 1:In its present form the manuscript requires significant background knowledge for full appreciation and would therefore mostly appeal to expert readers. Little additional information in particular pertaining to the choice of zebrafish strains could improve readability, e.g. the choice of flt1:YFP transgene in Figure 1A vs. kdrl:TagBFP in 1B vs dab2:GAL4FF; UAS:GFP in Suppl. Figure 1A to label the blood vessels, or the binary pdgfrb:GAL4FF; UAS:GFP system in 1E vs. pdgfrb:EGFP in 1H to label MCs.

We have adjusted in the text.

Figure 2 E: While ablation of the MCs can be followed by loss of the pdgfrb:GFP label, ablation and thereby treatment efficacy in the control area is not traceable by the presented approach. Furthermore, given the close association of MCs / aECs and the relatively poor axial focus of NIR light, a collateral damage of arterial vessels should be excluded e.g. by angiography as shown for toxin ablation in suppl. Figure 2B.

We have included the transmitted light images pre and post ablation to show un-disturbed flow in the artery as well as the wound side. See Supplementary movie S9 and S10, Supplementary Figure 2C

Figure 2H: Shouldn't the LEC migration distance be larger in control animals compared to the ablated situation? Are the labels mixed up?

Thank you for pointing this out, we have mixed up the labels while generating the illustrator file, it is fixed.

Suppl. Figure 2 A: The topic of high versus low NTR expression is only mentioned in passing and rather confusing. Can the authors provide additional information for the reader? Are these different fish lines?

We adjusted the text to eliminate the confusion.

Specific suggestions:Figure 2C: How many embryos is the bar diagram representing?

Information is now added to the figure legend.

Figure 2F: Legend appears mixed up, control is bottom, ablated group is on top.

This is fixed.

Figure 3F: The terms E3 water and PTU should be introduced and explained.

We have provided additional information in the legend and methods.

Figure 4G: It is not clear what the four groups refer to that are mentioned in the legend.

We have clarified the labelling in both Figure 4 G and F.

Figure 4J, model: What is indicated by the red asterisk, a reference in the legend is missing. It might be clearer to show a comparable sketch for the situation in the presence and absence of MCs.

We have included an explanation in the legend and we have modified the working model to include the WT situation.

There are language / text issues, carefully proofread the manuscript:Line 64: „…by 5 days post fertilization…" is redundant

This is fixed.

Line 65:.…, the vast majority of LECs is associated…

This is fixed.

Line 82: Studies using a vegfc reporter zebrafish line have uncovered multiple sources of vegfc, including the fibroblasts and neurons, which contribute to the initial sprouting

This is fixed.

Line 100: Thus, this study uncovers a close interaction between MC and LEC, which is of functional importance for lymphatic vessel formation in the zebrafish trunk…

This is fixed.

Line 163: However, in order to directly test if aEC function is critical for MC-dependent LEC migration in aISV

This is fixed.

Line 184:.…to avoid a possible contamination with cell types other than abcc9 positive MCs, we used

This is fixed.

Line 187: sentence appears incomplete – "…In line with published data that cxcr4a and cxcr4b are expressed by LECs and cxcl12b by aECs (Cha et al., 2012)…" alternatively delete that?

This is fixed.

Line 294: To quantify the migrating distance, the centre of PL nuclei in figure 3G and figure 4D was manually tracked until the cell either died or disappeared from the viewfield in Imaris.…

This is fixed.

Line 300: The migrating distance in all figures was measured in three-dimensions using the spot function in Imaris (Bitplane), see cell tracking

This is fixed.

Line 311: sentence appears incomplete: in a 2-well slide with separate chambers which allows.

This is fixed.

Line 440: laser ablation in 2 dpf Tg(flt1:YFP) (magenta); TgBAC(pdgfrb:GFP) (green) and Tg(5.2lyve1b:DsRed2) (grey) embryos?

This is fixed.

All supplementary movies run very fast, visibility could be improved by a slightly reduced frame rate.

We have slowed down the movies.

Reviewer 2:1. More details should be provided for Sup. Figure 2. What are HP, FP? How to use QTracker 705…etc.

We have fixed this accordingly.

References

Ando, K., Wang, W., Peng, D., Chiba, A., Lagendijk, A.K., Barske, L., Crump, J.G., Stainier, D.Y.R., Lendahl, U., Koltowska, K., et al. (2019). Peri-arterial specification of vascular mural cells from aive mesenchyme requires Notch signaling. Development *146*. 10.1242/dev.165589.

Harrison, M.R., Feng, X., Mo, G., Aguayo, A., Villafuerte, J., Yoshida, T., Pearson, C.A., Schulte-Merker, S., and Lien, C.L. (2019). Late developing cardiac lymphatic vasculature supports adult zebrafish heart function and regeneration. *ELife 8*. 10.7554/*eLife*.42762. Klein, Klara R., Karpinich, Natalie O., Espenschied, Scott T., Willcockson, Helen H., Dunworth, William P., Hoopes, Samantha L., Kushner, Erich J., Bautch, Victoria L., and Caron, Kathleen M. (2014). Decoy Receptor CXCR7 Modulates Adrenomedullin-Mediated Cardiac and Lymphatic Vascular Development. Developmental Cell *30*, 528-540. 10.1016/j.devcel.2014.07.012.

Matsuoka, R.L., Rossi, A., Stone, O.A., and Stainier, D.Y.R. (2017). CNS-resident progenitors direct the vascularization of neighboring tissues. Proc Natl Acad Sci U S A *114*, 10137-10142. 10.1073/pnas.1619300114.

Rajan, A.M., Ma, R.C., Kocha, K.M., Zhang, D.J., and Huang, P. (2020). Dual function of perivascular fibroblasts in vascular stabilization in zebrafish. PLoS Genet *16*, e1008800. 10.1371/journal.pgen.1008800.

Wang, G., Muhl, L., Padberg, Y., Dupont, L., Peterson-Maduro, J., Stehling, M., le Noble, F., Colige, A., Betsholtz, C., Schulte-Merker, S., and van Impel, A. (2020). Specific fibroblast subpopulations and neuronal structures provide local sources of Vegfc-processing components during zebrafish lymphangiogenesis. Nat Commun *11*, 2724. 10.1038/s41467020-16552-7.